# TraPO: A Semi-Supervised Reinforcement Learning Framework for Boosting LLM Reasoning

**Shenzhi Yang**[1][*]  **Guangcheng Zhu**[1][*]  **Haobo Wang**[1][†]  **Xing Zheng**[2]  **Yingfan Ma**[2]
**Zhongqi Chen**[2]  **Bowen Song**[2][†]  **Weiqiang Wang**[2]  **Junbo Zhao**[1]  **Gang Chen**[1]
[1] Zhejiang University   [2] Ant Group
{yangshenzhi,zhuguangcheng,wanghaobo,j.zhao,cg}@zju.edu.cn
{feishang.zx,bowen.sbw,weiqiang.wwq}@antgroup.com

## ABSTRACT

Reinforcement learning with verifiable rewards (RLVR) has proven effective in training large reasoning models (LRMs) by leveraging answer-verifiable signals to guide policy optimization, which, however, suffers from high annotation costs. To alleviate this problem, recent work has explored unsupervised RLVR methods that derive rewards solely from the model's internal consistency, such as through entropy and majority voting. While seemingly promising, these methods often suffer from model collapse in the later stages of training, which may arise from the reinforcement of incorrect reasoning patterns in the absence of external supervision. In this work, we investigate a novel semi-supervised RLVR paradigm that utilizes a small labeled set to *guide* RLVR training on unlabeled samples. Our key insight is that supervised rewards are essential for stabilizing consistency-based training on unlabeled samples, ensuring that only reasoning patterns verified on labeled instances are incorporated into RL training. Technically, we propose an effective policy optimization algorithm **TraPO** that identifies reliable unlabeled samples by matching their learning trajectory similarity to labeled ones. Building on this, TraPO achieves remarkable data efficiency and strong generalization on nine advanced benchmarks. With only 1K labeled and 3K unlabeled samples, TraPO reaches 42.6% average accuracy, surpassing the best unsupervised method trained on 45K unlabeled samples (38.3%). Notably, when using 4K labeled and 12K unlabeled samples, TraPO even *outperforms the fully supervised model* trained on the full 45K labeled samples on all benchmarks, while using only *10%* of the labeled data. The code is available via https://github.com/ShenzhiYang2000/TRAPO.

## 1 INTRODUCTION

The reinforcement learning with verifiable rewards (RLVR), pioneered by DeepSeek-R1 (Guo et al., 2025), has significantly advanced the development of large reasoning models (LRMs). In typical RLVR (Shao et al., 2024; Liu et al., 2025; Yu et al., 2025; Zheng et al., 2025), questions from a training corpus are fed into an LRM, which then generates multiple reasoning paths (rollouts) per input. Rewards are computed based on verifiable rules: most commonly, whether the final answer in a response matches the ground-truth label. By leveraging such an answer-verifiable structure, RLVR enables reward assignment through group-based advantage estimation, guiding the model to explore reasoning paths that lead to the correct final answer.

However, when scaling to large corpora, the reliance of this reward paradigm on gold-standard labels incurs prohibitively high annotation costs, making it difficult to generalize to specialized domains where ground-truth answers are scarce or expensive to obtain, such as medicine and finance (Wang et al., 2024b). To address this challenge, recent work has explored unsupervised RLVR methods

---

[*]Equal Contribution.  [†] Correspondence to Haobo Wang (wanghaobo@zju.edu.cn) and Bowen Song (bowen.sbw@antgroup.com)

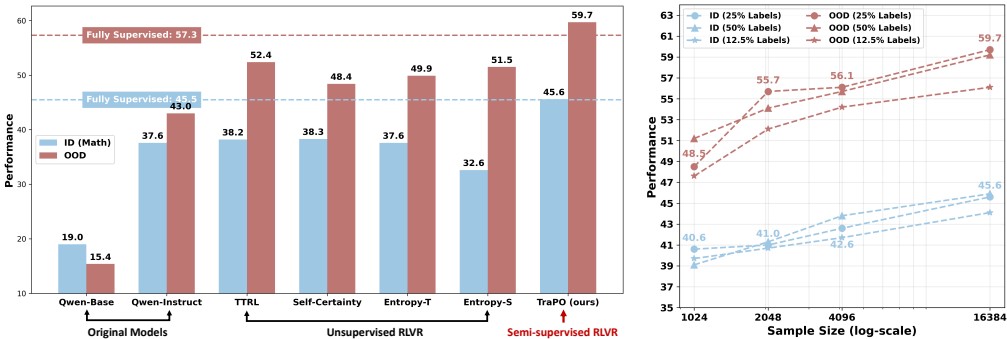

Figure 1: **Performance overview.** (Left) TRAPO surpasses fully supervised RLVR (45K samples) using just 10% (4K) annotated data. (Right) TRAPO scaling law: performance improves consistently with increasing sample sizes and varying annotation ratios. We only show the changes with a sample size at a 25% annotation rate in the figure; for other specific results, please see Table 12.

(Zhang et al., 2025a; Zhao et al., 2025; Agarwal et al., 2025; Li et al., 2025a; Zuo et al., 2025; Zhang et al., 2025a) that aim to eliminate dependence on external supervision directly. These approaches are grounded in the observation that LRMs have already internalized substantial knowledge during pretraining (Ye et al., 2025); thus, the goal shifts from learning factual correctness to eliciting latent reasoning capabilities through self-guided exploration. In this framework, rewards are computed based on intrinsic signals such as self-certainty (Zhao et al., 2025), entropy (Agarwal et al., 2025), or majority voting (Zuo et al., 2025), to encourage high-confidence and consistent outputs. Despite their promise, these unsupervised methods often fail to capture valid reasoning patterns and tend to reinforce incorrect consensus, leading to severe performance degradation in late training. This drawback can be attributed to the absence of external ground truth: the reward signal becomes self-reinforcing and prone to reinforcing systematic biases, leading to a degenerate feedback loop.

Analogous to human learning, unsupervised RLVR resembles a student solving problems based solely on current beliefs, treating the most confident answer as the ground truth. When incorrect, repeated reinforcement of the same reasoning path entrenches errors, leading to failure on both the current and related tasks. To break this vicious cycle, *humans typically learn from a few well-solved examples with verified solutions to establish a correct conceptual foundation*, then generalize via analogical reasoning. Therefore, we hypothesize that LRMs possess a similar property: a small number of verifiable labeled samples can enable LRMs to generalize patterns from larger amounts of unlabeled corpora. Inspired by this process, we propose a **Semi-supervised RLVR (SS-RLVR)** paradigm that takes advantage of a small set of labeled examples to anchor the reward signal, guiding the model toward reliable reasoning patterns and allowing more robust self-improvement.

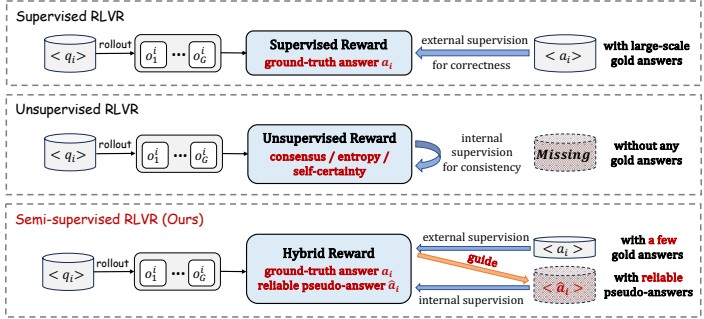

Figure 2: Different RLVR training paradigms.

Although promising in principle, our experiments show that simply combining supervised and unsupervised RLVR algorithms delivers only marginal benefits. For example, when combined with 3K entropy-based unlabeled RLVR training, the 1K supervised baseline only improves 0.6% accuracy. We argue that such failure stems from the neglect of internal links between labeled and unlabeled sets. In other words, only those reasoning patterns that are verified on labeled instances should be incorporated into RL training, and labeled data should be used as role models (Tarvainen & Valpola, 2017) to *guide* robust learning on unlabeled instances, as shown in Figure 2. Based on this key insight, we propose **TRAPO** (**Tra**jectory-based **P**olicy **O**ptimization), which measures the

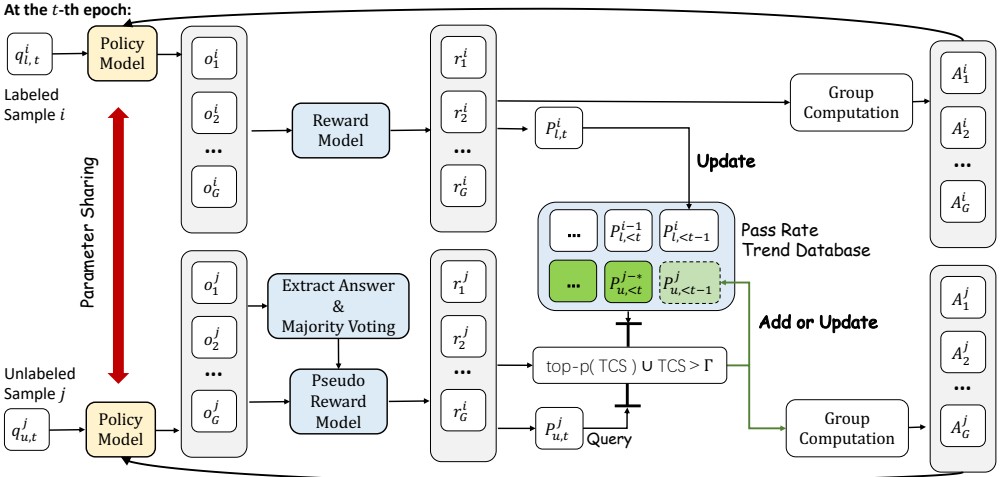

Figure 3: **TRAPO** is a semi-supervised RLVR training framework to dynamically select reliable unlabeled samples throughout the training process based on pass rate trajectory matching.

similarity between unlabeled and labeled samples in terms of their pass rate trajectories and uses this alignment as a criterion to select unlabeled samples with reliable pseudo-supervision for training. Experimental results demonstrate that TRAPO, trained with only 1K labeled and 3K unlabeled samples, achieves a $4.3\%$ improvement in in-domain performance over the strongest unsupervised baseline (trained on 45K unlabeled samples), $2.6\%$ over the best naive semi-supervised method, and $3.2\%$ over the supervised baseline (trained on 1K labeled samples). Notably, with 4K labeled and 12K unlabeled samples, TRAPO surpasses the fully supervised model trained on all 45K labeled samples across all benchmarks, using only 10% of the labeled data (see Figure 1, left). The scaling law for TRAPO (Figure 1, right) further demonstrates that with increased data and a labeling ratio (e.g, 25%), TRAPO achieves or approaches fully supervised performance without extra labels. These results strongly demonstrate TRAPO's ability to balance data efficiency and learning effectiveness.

## 2 RELATED WORK

**Semi-supervised Learning** leverages both labeled and unlabeled data to improve model performance, typically by exploiting data structure (Chapelle et al., 2009; Rasmus et al., 2015) or consistency assumptions (Laine & Aila, 2016; Berthelot et al., 2019; Xie et al., 2020; Sohn et al., 2020). In traditional classification tasks, outputs are drawn from a shared discrete label space, enabling effective label propagation via feature similarity. However, in RLVR, each input has an instance-specific solution space, where "correct" outputs vary significantly across examples. This makes direct alignment of unlabeled samples with labeled ones through standard similarity-based methods impractical, posing a key challenge in bridging labeled and unlabeled data for RLVR. Thus, in this paper, we turn from *what* the model learns to *how* it learns and employ the pass rate change trajectory as a medium to bridge the gap.

**Unsupervised RLVR** is built upon supervised RLVR, which has proven effective for aligning reasoning models in domains with executable or exact feedback, such as math and code (Hu et al., 2025; Guo et al., 2025; Shao et al., 2024), using deterministic, rule-based reward verifiers (Jaech et al., 2024). However, its reliance on outcome supervision limits applicability to tasks lacking a clear ground truth. Recent work explores unsupervised RLVR, which uses intrinsic, self-generated signals to enable reward-free training. Methods include self-rewarding via judgment prompting (Wu et al., 2024; Yuan et al., 2024; Xiong et al., 2025) or ensemble heads (Wang et al., 2024c; Zhou et al., 2025), though often costly for online use. More scalable approaches leverage lightweight signals—such as entropy (Agarwal et al., 2025), self-confidence (Li et al., 2025a), or majority voting (Zuo et al., 2025)—to guide online policy updates (Zhang et al., 2025a; Zhao et al., 2025). However, purely unsupervised training risks model collapse due to biased or noisy signals rein-

forcing incorrect behaviors (Zhang et al., 2025c;b). Our work builds on this line by introducing a new semi-supervised framework that anchors learning with labeled data to correct intrinsic signals, improving stability and generalization.

**Reasoning Data Selection** is a critical step in training LRMs, which can be broadly categorized into external and internal approaches. External methods rely on auxiliary resources such as human annotations (Li et al., 2022), knowledge bases (Nguyen et al., 2024), or proxy models (He et al., 2025a) to evaluate correctness and confidence, but suffer from limited applicability due to dependency on external resources (Bi et al., 2025). In contrast, internal methods leverage model-internal signals, such as output probabilities (Plaut et al., 2024), semantic entropy (Kuhn et al., 2023), hidden representations (Wang et al., 2024a), or the changes in reward (Li et al., 2025b) to estimate data quality in a label-free manner. Nevertheless, such metrics do not reflect the fundamental characteristics of data that are most beneficial for model learning. In this work, we go beyond superficial indicators by probing the intrinsic learning dynamics of the data, thereby identifying unlabeled instances that genuinely contribute to effective and robust model training.

## 3 METHOD

In this section, we present our semi-supervised reinforcement learning paradigm, which uses limited labeled data to guide reliable policy learning on large-scale unlabeled data. In Section 3.1, we discuss the limitations of supervised and unsupervised RLVR, and highlight the motivation for semi-supervised RLVR. In Section 3.2, we explore the bridge between labeled and unlabeled data, propose a trajectory-based method to select reliable rewards and provide theoretical analysis on generalization.

### 3.1 SEMI-SUPERVISED REINFORCEMENT LEARNING WITH VERIFIABLE REWARDS

**Supervised RLVR.** In traditional RLVR, we assume access to a large labeled dataset $\mathcal{D}_l = \{(q_i, y_i)\}_{i=1}^{N_l}$, where each sample consists of a question $q_i$ and its corresponding verifiable ground-truth answer $y_i$. For each question $q_i$, we input it into a policy model $\pi_\theta$ to generate $G$ candidate outputs, denoted as $\{\tau_i^j\}_{j=1}^{G}$. Given the ground-truth answer $y_i$ as a supervision, we assign rewards to the generated responses based on whether they derive the correct answer. Specifically, we define a binary reward function that evaluates the final extracted answer from each output $\tau_i^j$:

$$R(\tau_i^j, y_i) = \mathbb{I}(\tau_i^j, y_i) = \begin{cases} 1 & \text{if } a_i^j = y_i, \\ 0 & \text{otherwise.} \end{cases} \tag{1}$$

Here, $a_i^j = \texttt{extract}(\tau_i^j)$ denotes the answer extracted from the generated response $\tau_i^j$, such as the content within boxed delimiters (e.g., \boxed{·}). With the ground-truth answers $y_i$ serving as explicit guidance signals, this *Supervised RLVR* paradigm reinforces only the responses that yield the correct answers; the policy model $\pi_\theta$ is gradually steered toward discovering valid and consistent reasoning paths, thereby enabling stable and scalable policy optimization.

**Unsupervised RLVR.** Although supervised RLVR has achieved great success, its reliance on golden answers $y_i$ incurs high annotation costs. To address this, the community has explored unsupervised RLVR techniques that rely solely on unlabeled data $\mathcal{D}_u = \{q_i\}_{i=1}^{N_u}$. Under this setting, the absence of golden answers necessitates the use of proxy rewards $R_u(\tau_i^j)$ that estimate $R(\tau_i^j, y_i)$ based on the model's confidence or consensus $\texttt{conf}(·)$. A widely adopted method is majority voting, where the reward is defined as:

$$R_u(\tau_i^j) = \texttt{conf}(\pi_\theta(\tau_i^j \mid q_i)) = \mathbb{I}(a_i^j = \text{MAJ}(a_i^1, a_i^2, \cdots, a_i^G)) \tag{2}$$

where MAJ(·) denotes the pseudo-label $\tilde{y}$ obtained by majority answer among $G$ rollouts. This approach effectively treats the most frequently generated answer as the pseudo-label, providing a form of self-supervised signal. Beyond majority voting, Zhao et al. (2025) use self-certainty, Agarwal et al. (2025) use token-level or sequence-level entropy as a proxy for confidence, and compute rewards accordingly. Fundamentally, these methods are based on a key assumption: higher confidence implies a greater probability of producing the correct answer, and thus the higher the reward it should receive.

However, this assumption breaks down when the proxy reward diverges from actual correctness. Take the majority voting as an example, if the majority answer is not the correct answer, *i.e.*, $\text{MAJ}(a_i^1, \cdots, a_i^G) \neq y_i$, then the incorrect responses are reinforced. This creates a dangerous feedback loop: the policy becomes more confident in the wrong answer, leading to even stronger wrong consensus in subsequent iterations. Over time, the model converges to a state where it confidently produces incorrect outputs.

**Semi-supervised RLVR.** To break this vicious loop induced by the absence of grounded feedback, we hypothesize that we must introduce labeled examples to anchor the reward to ground truth. Formally, we adopt a hybrid reward function that computes rewards differently for labeled and unlabeled data:

$$R_{\text{semi}}(\tau_i^j) = \begin{cases} R(\tau_i^j, y_i), & \text{if } (q_i, y_i) \in \mathcal{D}_l, \\ R_u(\tau_i^j), & \text{if } q_i \in \mathcal{D}_u. \end{cases} \tag{3}$$

Here, labeled data are used to compute rewards under supervision from the ground-truth labels $y_i$, while unlabeled data can adopt *any* self-consistency-based reward we have stated previously. Since the reward $R(\tau_i^j, y_i)$ of labeled data is independent of the model's consensus, this training paradigm introduces a crucial distinction between correctness (alignment with ground truth) and self-consistency (internal agreement among outputs), thereby preventing the policy from reinforcing incorrect but internally consistent outputs.

The design of our Semi-supervised RLVR framework stems from the inherent trade-off between *data efficiency* and *learning effectiveness*. Compared to unsupervised variants, SS-RLVR effectively guides robust learning on unlabeled instances by using labeled data as a reliable anchor. In contrast to fully supervised approaches, it significantly reduces the need for costly annotation—our experiments show that SS-RLVR achieves performance close to supervised learning using only **25%** of the labeled data. In practice, this trade-off not only directly reduces the annotation burden, but also enables high-quality data synthesis within iterative refinement pipelines, thereby improving data quality over time. This makes SS-RLVR particularly attractive for domains where labeled data is scarce or expensive to obtain, such as medicine and finance.

## 3.2 Progressive Trajectory Guidance for Bridging Labeled and Unlabeled Data

Despite its promise, we show that a trivial baseline that simply combines supervised and unsupervised RLVR algorithms delivers only marginal benefits. For example, when supplemented with 3K entropy-based unlabeled RLVR training, the 1K supervised baseline achieves merely a 0.6% accuracy improvement. This suggests that such a naive strategy remains constrained by the internal signals of LRMs and suffers from the internal ungrounded reasoning patterns. Thus, SS-RLVR must move beyond shallow integration and instead uncover the deeper intrinsic relationships between labeled and unlabeled data. In particular, the key is to exploit those reasoning patterns in unlabeled data that can be externally validated by labeled examples. To achieve this goal, it is required to identify a shared, meaningful signal that transcends the heterogeneity of solution spaces and reliably reflects the model's ability to transfer knowledge from labeled to unlabeled data.

In this work, we propose **TRAPO** (**Tra**jectory-based **P**olicy **O**ptimization), which leverages the learning dynamics of LRMs across training steps as a proxy to connect labeled and unlabeled data, as shown in Figure 3. Specifically, at each step $t$, TRAPO computes the pass rate for each training point. We then identify those unlabeled samples whose *pass rate trajectories* closely align with those of labeled samples as reliable data, which means that their reasoning patterns can be externally validated by the labeled set. In other words, we hypothesize that when an unlabeled sample is well-learned, its pass rate trajectory should exhibit trends consistent with those observed in labeled data. Naturally, since pass rates cannot be directly computed for unlabeled data, we introduce a pseudo–pass rate approximation to serve as a proxy. Formally, for a question $q$ at epoch $t$, the (pseudo) pass rate is defined as the fraction of generated responses that satisfy the expected answer criteria:

$$P_q^{(t)} = \begin{cases} \frac{1}{G} \sum_{i=1}^{G} \mathbb{I}(a_i^{(t)} = \tilde{y}^{(t)}), & q \in \mathcal{D}_u, \\ \frac{1}{G} \sum_{i=1}^{G} \mathbb{I}(a_i^{(t)} = y), & q \in \mathcal{D}_l, \end{cases} \tag{4}$$

Then, we define the *pass rate trajectory* of question $q$ as the sequence of its pass rates across training epochs:

$$\mathbf{T}_q^{(t)} = \left[ P_q^{(1)}, P_q^{(2)}, \ldots, P_q^{(t)} \right] \in [0,1]^t, \tag{5}$$

initialized as $\mathbf{T}_q^{(0)} = [\,]$ and updated iteratively via concatenation: $\mathbf{T}_q^{(t)} = \mathbf{T}_q^{(t-1)} \oplus P_q^{(t)}$, where $\oplus$ denotes sequence concatenation. We maintain a reliable pass rate database $\mathcal{D}_{\text{reliable}}$, initialized with all labeled sample trajectories: $\mathcal{D}_{\text{reliable}}^{(0)} = \{ \mathbf{T}_l \mid l \in \mathcal{D}_l \}$. Reliably pseudo-labeled trajectories from unlabeled data selected in subsequent steps are added to update this database. The average trajectory of this database, $\bar{\mathbf{T}}_{\text{reliable}}^{(t)} = \frac{1}{|\mathcal{D}_{\text{reliable}}|} \sum_{\mathbf{T} \in \mathcal{D}_{\text{reliable}}} \mathbf{T}$, serves as a trusted reference for assessing the reliability of unlabeled samples based on trajectory alignment. Then we compute a trajectory-based cosine similarity (TCS) as:

$$\text{TCS}(\mathbf{T}_u^{(t)}, \bar{\mathbf{T}}_{\text{reliable}}^{(t)}) = \hat{\mathbf{T}}_u^{(t)} \cdot \hat{\bar{\mathbf{T}}}_{\text{reliable}}^{(t)} = \sum_{j=1}^{t} \hat{P}_u^{(j)} \cdot \hat{\bar{P}}_{\text{reliable}}^{(j)} \tag{6}$$

where $\hat{P}_u^{(j)} = \frac{P_u^{(j)}}{\sqrt{\sum_{i=1}^{t}(P_u^{(i)})^2}}$ and $\hat{\bar{P}}_{\text{reliable}}^{(j)} = \frac{\bar{P}_{\text{reliable}}^{(j)}}{\sqrt{\sum_{i=1}^{t}(\bar{P}_{\text{reliable}}^{(i)})^2}}$ are the normalized pass rate of the unlabeled sample and the reliable database, respectively.

To select the reliable trajectories, we combine two criteria: the `top-p` of unlabeled samples with highest trajectory similarity to the labeled data, and any sample whose similarity exceeds a threshold $\Gamma$.

$$\mathbf{M}(u) = \mathbb{I}\left( u \in \texttt{top-p}\left( \text{TCS}(\mathbf{T}_u, \bar{\mathbf{T}}_{\text{reliable}}) \right) \right) \vee \mathbb{I}\left( \text{TCS}(\mathbf{T}_u, \bar{\mathbf{T}}_{\text{reliable}}) \geq \Gamma \right) \tag{7}$$

With this selection mask in hand, we now integrate it into the training process to ensure only reliably improving samples influence model updates. To ensure stability, we employ a warm-up phase using only labeled data for updates, while accumulating unlabeled trajectories. After warm-up, we apply the mask M to include only reliable unlabeled samples:

$$\mathcal{L}(\theta) = \mathcal{J}_{\text{GRPO}}^{\text{labeled}}(\theta) + \mathbf{M} \odot \mathcal{J}_{\text{GRPO}}^{\text{unlabeled}}(\theta). \tag{8}$$

where $\odot$ denotes the dot product of vectors. Here, $\mathcal{J}_{\text{GRPO}}$ is the GRPO objective (Shao et al., 2024):

$$\mathcal{J}_{\text{GRPO}}(\theta) = \frac{1}{\sum_{i=1}^{G} |\tau_i|} \sum_{i=1}^{G} \sum_{l=1}^{|\tau_i|} \text{CLIP}(\gamma_{i,l}(\theta), A_i, \epsilon) - \beta \cdot \mathbb{D}_{\text{KL}}[\pi_\theta \| \pi_{\text{ref}}] \tag{9}$$

where $\gamma_{i,l}(\theta) = \pi_\theta(\tau_{i,l}|q, \tau_{i,<l}) / \pi_{\theta_{\text{old}}}(\tau_{i,l}|q, \tau_{i,<l})$ is the importance sampling term, and $\text{CLIP}(\gamma, A, \epsilon) = \min[r \cdot A, \text{clip}(\gamma; 1 - \epsilon, 1 + \epsilon) \cdot A]$ is the clipped surrogate objective.

In summary, we propose leveraging the evolution of correctness during training (*pass rate trajectories*) as a reliable signal for evaluating unlabeled samples. By measuring the similarity between the pass rate trajectory of an unlabeled instance and the average trajectory derived from labeled data, we identify samples whose learning dynamics align closely with those observed under trusted supervision. To validate the effectiveness of TRAPO in selecting high-quality unlabeled samples and grounding unsupervised learning within a stable feedback framework, we provide a theoretical analysis of its generalization error bound:

---

**Theorem 3.1** (Trajectory-Consistent Generalization). *(Informal) Let the generalization error of policy $\pi_\theta^{(t)}$ be the expected risk on the true distribution. Assuming $L_y$ is the label space diameter, under the* TRAPO *framework, with probability at least $1 - \delta$, this error is bounded by:*

$$\mathcal{R}_{\mathcal{D}_l}(\pi_\theta^{(t)}) + \lambda' + \alpha \cdot \mathbb{E}_{q' \sim \mathcal{D}_u}\left[ 1 - \textit{TCS}(\mathbf{T}_{q'}^{(t)}, \bar{\mathbf{T}}_{\text{reliable}}^{(t)}) \right] + L_y \left( 1 - \bar{C}^{(t)} + \sqrt{\frac{\ln(2n/\delta)}{2G}} \right) \tag{10}$$

*where $\mathcal{R}_{\mathcal{D}_l}(\pi_\theta^{(t)})$ is the empirical risk on $\mathcal{D}_l$, $\lambda' = \lambda + \lambda_d \geq 0$ bounds the domain shift between $\mathcal{D}_l$ and $\mathcal{D}_u$, and $\bar{C}^{(t)}$ is the average voting confidence across $n$ samples based on $G$ votes.*

---

Table 1: Overall performance based on Qwen2.5-Math-7B under three different training paradigms. **Bold** and underline indicate the best and second-best results, respectively.

| Model | In-Distribution Performance | | | | | | Out-of-Distribution Performance | | | |
|---|---|---|---|---|---|---|---|---|---|---|
| | AIME 24/25 | AMC | MATH-500 | Minerva | Olympiad | Avg. | ARC-c | GPQA* | MMLU-Pro | Avg. |
| *Original Models* | | | | | | | | | | |
| Qwen-Base | 11.5/4.9 | 31.3 | 43.6 | 7.4 | 15.6 | 19.0 | 18.2 | 11.1 | 16.9 | 15.4 |
| Qwen-Instruct | 12.5/10.2 | 48.5 | 80.4 | 32.7 | 41.0 | 37.6 | 70.3 | 24.7 | 34.1 | 43.0 |
| *Unsupervised Methods Trained on 45K Samples w/o Any Labels* | | | | | | | | | | |
| TTRL | 14.1/12.7 | 51.5 | 76.6 | 33.8 | 40.3 | 38.2 | 80.5 | 35.4 | 41.3 | 52.4 |
| Self-certainty | 16.9/10.2 | 51.7 | 77.6 | 34.9 | 38.8 | 38.3 | 72.9 | 30.8 | 41.4 | 48.4 |
| Token-level Entropy | 15.0/9.9 | 50.3 | 75.2 | 36.8 | 38.4 | 37.6 | 75.6 | 33.3 | 40.9 | 49.9 |
| Sentence-level Entropy | 11.4/10.7 | 42.1 | 68.0 | 32.7 | 30.5 | 32.6 | 79.4 | 32.3 | 42.7 | 51.5 |
| *Semi-supervised Methods Trained on 1K Labeled Samples & 3K Unlabeled Samples* | | | | | | | | | | |
| Fully Supervised w/ *1K* Labels | 14.2/13.5 | 52.6 | 80.2 | 34.9 | 40.9 | 39.4 | 76.2 | 36.4 | 43.6 | 52.1 |
| TTRL | 14.9/10.7 | 55.3 | 77.8 | 33.1 | 43.6 | 39.2 | 72.6 | 35.4 | 42.7 | 50.2 |
| Self-certainty | 16.5/11.4 | 55.6 | 79.8 | 35.3 | 41.2 | 40.0 | 64.8 | 30.3 | 41.6 | 45.6 |
| Token-level Entropy | 18.2/11.9 | 53.4 | 80.2 | 34.6 | 41.9 | 40.0 | 72.9 | 32.3 | 44.0 | 49.7 |
| Sentence-level Entropy | 15.4/11.5 | 54.9 | 79.4 | 36.0 | 41.2 | 39.7 | 79.4 | 33.8 | 44.5 | 52.6 |
| **TRAPO (ours)** | 17.9/13.8 | 58.7 | 81.4 | 38.2 | 45.5 | 42.6 | 83.7 | 37.9 | 46.8 | 56.1 |
| Fully Supervised w/ *4K* Labels | 19.6/14.8 | 57.9 | 80.6 | 39.3 | 46.5 | 43.1 | 82.1 | 39.9 | 48.2 | 56.7 |
| *TRAPO Trained on 4K Labeled Samples & 12K Unlabeled Samples* | | | | | | | | | | |
| **TRAPO (ours)** | 24.3/17.1 | 60.0 | 84.6 | 39.3 | 48.3 | 45.6 | 84.6 | 43.9 | 50.7 | 59.7 |
| Fully Supervised w/ *45K* Labels | 25.1/15.3 | 62.0 | 84.4 | 39.3 | 46.8 | 45.5 | 82.3 | 40.4 | 49.3 | 57.3 |

Theorem 3.1 highlights the role of trajectory consistency as a regularizer in semi-supervised policy learning. Specifically, the term $\mathbb{E}_{q' \sim \mathcal{D}_u}\left[1 - \text{TCS}\big(\mathbf{T}_{q'}^{(t)}, \bar{\mathbf{T}}_{\text{reliable}}^{(t)}\big)\right]$ encourages unlabeled samples to follow learning dynamics similar to those of labeled data, effectively anchoring the optimization path. The dependence on $\bar{C}^{(t)}$ reflects the model's self-confidence during training, with lower confidence leading to a looser bound, thus promoting cautious updates. The formal theorem and its proof are presented in Appendix B.13.

## 4 EXPERIMENT

This section reports the main experimental results. Appendix E.1 compares more fully supervised baselines; E.2 further validates TRAPO on more models; E.3 shows that TRAPO is plug-and-play; E.4 evaluates TRAPO on the DeepMath dataset; E.5 compares TRAPO with other selection strategies; E.6 confirms TRAPO's stability; E.7 analyzes the temporal efficiency of TRAPO; E.8 examines different strategies for selecting utilization trajectories.

### 4.1 SETUP

**Dataset and Benchmarks.** We follow prior work Yan et al. (2025) and use the widely used math reasoning dataset OpenR1-Math-220k (Face, 2025) for training. For evaluation, we focus on six in-distribution (ID) math reasoning benchmarks: AIME 2024, AIME 2025, AMC (Li et al., 2024), Minerva (Lewkowycz et al., 2022), OlympiadBench (He et al., 2024), and MATH-500 (Hendrycks et al., 2021). We report `avg@32` on AIME 2024/2025 and AMC (due to small test sets) and `pass@1` on the others. For out-of-distribution (OOD) generalization, we evaluate on ARC-c (Clark et al., 2018), GPQA-diamond (Rein et al., 2024) (GPQA*), and MMLU-Pro (Wang et al., 2024b), covering open-domain reasoning, graduate-level science, and academic reasoning. All evaluations use temperature sampling with $T = 0.6$.

**Baseline Methods.** We evaluate supervised, unsupervised, and semi-supervised RLVR methods across varying data scales. For supervised training, we apply GRPO on 1K, 4K, and 45K labeled samples. In the unsupervised setting, we remove ground-truth labels from the full 45K dataset and evaluate four approaches: (1) **TTRL** (Zuo et al., 2025), which uses majority-voted outputs

Table 2: Performance of different training paradigms with 1K labeled math (ID) samples and 1K unlabeled non-math (OOD) samples. **Bold** and underline indicate the best and second-best results, respectively.

| Model | In-Distribution Performance | | | | | | Out-of-Distribution Performance | | | |
| --- | --- | --- | --- | --- | --- | --- | --- | --- | --- | --- |
| | AIME 24/25 | AMC | MATH-500 | Minerva | Olympiad | Avg. | ARC-c | GPQA* | MMLU-Pro | Avg. |
| Original Model | | | | | | | | | | |
| Qwen-Base | 11.5/4.9 | 31.3 | 43.6 | 7.4 | 15.6 | 19.0 | 18.2 | 11.1 | 16.9 | 15.4 |
| Qwen-Instruct | 12.5/10.2 | 48.5 | 80.4 | 32.7 | 41.0 | 37.6 | 70.3 | 24.7 | 34.1 | 43.0 |
| Unsupervised Methods Trained on *1K* Unlabeled ID Samples & *1K* Unlabeled OOD Samples | | | | | | | | | | |
| TTRL | 13.3/9.4 | 48.2 | 72.2 | 27.6 | 34.8 | 34.3 | 76.7 | 33.8 | 36.2 | 48.9 |
| Self-certainty | 18.5/9.6 | 53.4 | 79.6 | 33.4 | 40.4 | 39.2 | 76.7 | 37.9 | 45.6 | 53.4 |
| Token-level Entropy | 14.6/13.3 | 46.8 | 77.6 | 27.9 | 40.1 | 36.7 | 74.5 | 36.4 | 35.8 | 48.9 |
| Sentence-level Entropy | 16.4/11.5 | 51.8 | 74.0 | 33.5 | 37.2 | 37.4 | 74.5 | 34.8 | 43.3 | 50.9 |
| Semi-supervised Methods Trained on *1K* Labeled ID Samples & *1K* Unlabeled OOD Samples | | | | | | | | | | |
| TTRL | 16.4/13.6 | 49.9 | 66.9 | 26.5 | 37.8 | 35.2 | 62.0 | 31.8 | 43.5 | 45.8 |
| Self-certainty | 16.0/10.9 | 53.0 | 78.4 | 34.2 | 39.0 | 38.6 | 77.1 | 32.8 | 45.7 | 51.9 |
| Token-level Entropy | 17.7/11.0 | 51.7 | 77.0 | 33.1 | 41.0 | 38.6 | 76.5 | 30.8 | 44.7 | 50.7 |
| Sentence-level Entropy | 15.7/10.0 | 51.4 | 77.4 | 34.9 | 37.5 | 37.8 | 75.1 | 31.3 | 44.3 | 50.2 |
| **TRAPO (ours)** | 18.5/15.7 | 53.4 | 80.4 | 33.8 | 44.0 | **41.0** | 83.6 | 38.9 | 48.1 | **56.9** |
| Fully Supervised w/ *2K* Labels | 17.3/12.4 | 56.8 | 81.4 | 38.6 | 44.8 | 41.9 | 82.0 | 38.9 | 52.4 | 57.8 |

as pseudo-labels; (2) **Self-Certainty** (Zhao et al., 2025), which maximizes KL divergence to encourage confident predictions; (3) **Token-Level Entropy** (Agarwal et al., 2025), which minimizes token-level entropy for consistency; and (4) **Sentence-Level Entropy** (Agarwal et al., 2025), which maximizes sentence likelihood. For semi-supervised training, we use 1K labeled and 3K unlabeled samples, applying GRPO on the labeled subset and each unsupervised method on the unlabeled subset to form hybrid baselines. We further evaluate a stronger setting with 4K labeled and 12K unlabeled samples to assess performance under higher label efficiency. In E.1, we compare with more supervised baselines (Zeng et al., 2025b; Hu et al., 2025; Cui et al., 2025; Liu et al., 2025).

## 4.2 EXPERIMENTAL RESULTS

**TRAPO achieves SOTA performance.** Our main results are summarized in Table 1. First, TRAPO significantly outperforms all fully unsupervised baselines using only 1K labeled samples (with 3K unlabeled). Compared to the best unsupervised method trained on the full 45K unlabeled set, TRAPO achieves gains of $4.3\%$ in ID and $3.7\%$ in OOD accuracy, demonstrating that even minimal labeled data can lead to substantial improvements when effectively integrated. Second, TRAPO outperforms naive semi-supervised approaches that treat labeled and unlabeled data independently, improving the strongest such baseline by $2.6\%$ (ID) and $3.5\%$ (OOD), which underscores the importance of using labels to actively guide the learning from unlabeled examples. Finally, TRAPO surpasses the fully supervised model trained on the same 1K labels by $3.2\%$ (ID) and $4.0\%$ (OOD). It matches the performance of a fully supervised model trained on 4K labels while using only 25% of the labeled data. Notably, when trained with 4K labeled and 12K unlabeled samples, TRAPO achieves 45.6 ID and 59.7 OOD accuracy, exceeding the fully supervised model trained on all 45K labels by $0.1\%$ (ID) and $2.4\%$ (OOD), despite using only *10%* of the total labels. This remarkable performance highlights TRAPO's superior data efficiency and generalization capability.

**TRAPO succeeds with OOD unlabeled data.** To investigate whether labeled data can guide learning on out-of-domain (OOD) unlabeled data, we evaluate a semi-supervised setup with 1K labeled samples from the *mathematics* domain (ID) and 1K unlabeled samples from *non-mathematical* domains (OOD). This cross-domain setting is challenging due to the limited transfer of reasoning patterns across domains. As shown in Table 2, naive semi-supervised methods fail to benefit from labeled data well. For instance, self-certainty drops by $0.6\%$ on ID and $1.5\%$ on OOD, indicating that naive integration of labeled and unlabeled data harms learning under domain shift. In contrast, TRAPO achieves significant improvements, outperforming the best unsupervised baseline by $1.8\%$ on ID and $3.5\%$ on OOD. It also closely matches the fully supervised model with 2K labels,

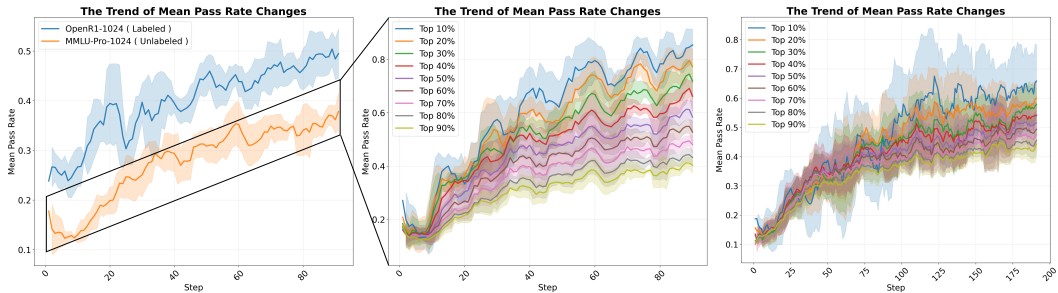

Figure 4: Left: Average performance changes on labeled and unlabeled data. Center: Unlabeled data performance vs. trajectory matching score using **true** training dynamics on unlabeled data. Right: Unlabeled data performance vs. trajectory matching score using **pseudo** training dynamics on unlabeled data.

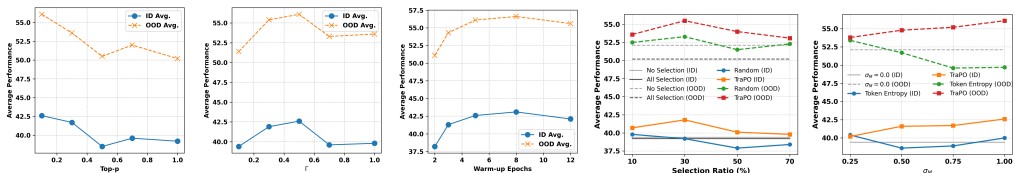

Figure 5: Sensitivity Analysis. The left three plots show sensitivity analyses of top-p, $\Gamma$, and warmup epochs (Tables 9, 10, and 11 in the Appendix). The right two plots compare performance for different ratios of selected and available unlabeled samples ($3K \times \sigma_M$). See tables 14 and 15 in the Appendix for details.

trailing by only $0.9\%$ on both metrics. The substantial gain in OOD performance demonstrates that TRAPO enables robust cross-domain generalization, highlighting its strong ability to transfer reasoning knowledge even under domain discrepancy.

**Effectiveness of trajectory matching.** To evaluate whether trajectory matching identifies reliable unlabeled examples, we analyze the link between trajectory similarity and performance. As shown in the middle plot of Figure 4, samples with dynamics more aligned to labeled data achieve much higher performance. The top 10% of samples outperform the bottom 10% by over **40**%, confirming that alignment correlates with reliability. In practice, we use pseudo-labels from voting to estimate unlabeled sample dynamics. The right plot of Figure 4 shows that matching pseudo dynamics to true labeled dynamics still yields a strong positive correlation with final performance. This validates the practical utility of TRAPO.

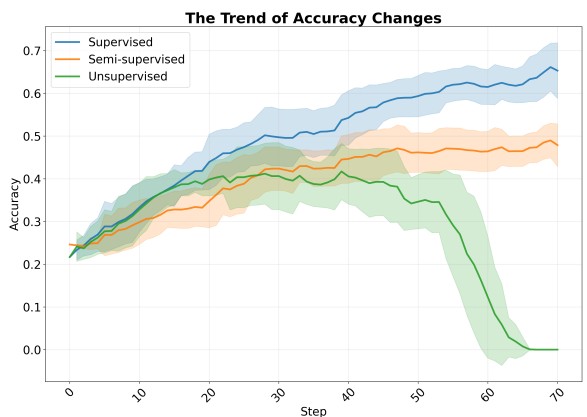

Figure 6: Performance comparison on Llama-3.1-8B.

**Sensitivity analysis.** We systematically analyze the impact of top-p, $\Gamma$, and warm-up length with the Qwen-2.5-7B model using 1K labeled and 3K unlabeled samples (left three plots in Figure 5). For top-p, larger values lead to noisy early-stage predictions and unreliable pseudo-labels, degrading overall performance. For $\Gamma$, setting it too low admits too many low-quality unlabeled samples, while setting it too high is overly conservative, leading to underutilization; both extremes harm the model. Short warm-up lengths lead to unstable pseudo-labeling, but performance stabilizes as the warm-up lengthens. With different selection ratios and varying proportions ($\sigma_M$) of available unlabeled samples, TRAPO outperforms random selection and a strong token-level entropy baseline (the right two plots in Figure 5). We find

that TRAPO achieves optimal results using the top 30% of unlabeled samples, whereas adding more unlabeled samples increases noise and reduces gains. These experiments highlight the critical role of intelligent denoising and selection strategies.

**Experiments with other LLMs.** Besides Qwen, we also compare the training effectiveness of the three paradigms using the Llama-3.1-8B-Instruct model. The model performance during training is shown in Figure 6, and detailed results are presented in Table 5. Here, our semi-supervised TRAPO method exhibits a similar trend to supervised training and maintains consistent improvement. In contrast, unsupervised training leads to a rapid performance collapse within tens of training steps. This underscores the critical importance of effective pseudo-supervision selection via trajectory matching in stabilizing the training process.

## 5 CONCLUSION

In this paper, we present the first exploration of semi-supervised learning in the RLVR setting. We introduce a novel paradigm that leverages a small set of labeled data to guide robust self-improvement on unlabeled data. We propose TRAPO (Trajectory based Policy Optimization), a method that enables reliable pseudo-supervision by aligning the learning dynamics of labeled and unlabeled samples through trajectory similarity in pass rate progression. Results show TRAPO significantly outperforms various baselines using only a fraction of labeled data, achieving an exceptional balance between efficiency and effectiveness.

## 6 ACKNOWLEDGEMENT

This paper was supported by the NSFC under Grants (No. 62402424) and Ant Group through CCF-Ant Research Fund (No. CCF-AFSG RF20240406). This paper was also supported by Fundamental and Interdisciplinary Disciplines Breakthrough Plan of the Ministry of Education of China.

## 7 REPRODUCIBILITY STATEMENT

We are committed to ensuring the reproducibility of our work. To this end, we will open-source code, model weights, and processed datasets upon paper acceptance. The codebase will include detailed documentation and training scripts to reproduce experimental results reported in the paper.

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

# Appendix

## A  LLM USAGE

In the preparation of this paper, the LLM was used solely for language editing and proofreading to improve clarity and readability.

Table 3: Table of Notations and Descriptions

| Notation | Description |
|---|---|
| **Optimization and Reward Setup** | |
| $\mathcal{J}$ | Group Relative Policy Optimization (GRPO): policy update via response grouping and relative advantage. |
| $r_i \in \{0, 1\}$ | Binary reward: 1 for correct, 0 for incorrect response. |
| $\mathcal{J}_{\text{pref}}$ | Equivalent preference optimization objective under binary rewards. |
| $p$ | Empirical accuracy: fraction of correct responses in a batch. |
| $N^+, N^-$ | Expected number of correct and incorrect responses: $N^+ = pN$, $N^- = (1-p)N$. |
| $p^+, p^-$ | Group-specific weights: $p^+ = \frac{1-p}{\sqrt{p(1-p)}}, p^- = \frac{p}{\sqrt{p(1-p)}}$. |
| $\hat{A}_{i,l}$ | Advantage estimator: $\hat{A}_{i,l} = \frac{r_i - p}{\sqrt{p(1-p)}}$. |
| $r_{i,l}(\theta)$ | Probability ratio between current and old policy for token generation. |
| $\text{clip}(\cdot, 1 \pm \varepsilon)$ | Clipping function to stabilize policy updates. |
| **Generalization and NTK Analysis** | |
| $\Delta \log \pi^t(\tau_k' \| q')$ | Change in log-probability of response $\tau_k'$ after update. |
| $\Theta((q, \tau), (q', \tau'))$ | Response-level NTK: $\langle \nabla_\theta \log \pi(\tau\|q), \nabla_\theta \log \pi(\tau'\|q') \rangle$. |
| $\Theta_{++} > 0, \Theta_{--} > 0$ | Gradient alignment: correct-correct and error-error responses align. |
| Orthogonal gradients | Correct and incorrect response gradients are orthogonal. |
| $D_{\text{traj}}^{(t)}(q, q')$ | Trajectory divergence: $1 - \cos \angle$ between response pass rate. |
| $\text{sign}(\Delta \log \pi^t) = +1$ | Positive generalization: similar questions benefit from training. |
| **Convergence and Risk Bounds** | |
| $d_{\mathcal{H}\Delta\mathcal{H}}(\mathcal{D}_l, \mathcal{D}_u)$ | Domain discrepancy: maximum distinguishability under $\mathcal{H}$. |
| $d_{\mathcal{H}\Delta\mathcal{H}} \leq \alpha \mathbb{E}[D_{\text{traj}}] + \lambda_d$ | Trajectory divergence bounds domain shift. |
| $R_{\mathcal{D}_u}(\pi_\theta^{(t)})$ | Generalization risk on target domain. |
| $\mathcal{R}_{TC}^{(t)}$ | Dynamic trajectory consistency risk: $\alpha \mathbb{E}[D_{\text{traj}}^{(t)}] + L_y(1 - \bar{C}^{(t)})$. |
| $\bar{C}^{(t)}$ | Average confidence (e.g., pass rate) at iteration $t$. |
| $U_t = \mathbb{E}[R_{\mathcal{D}_u}(\pi_\theta^{(t)})]$ | Expected target risk, used in convergence analysis. |
| $U_{t+1} \leq U_t - \eta_t \xi_t + \beta_t$ | Monotonic convergence inequality under consistent learning. |
| $\beta_t$ | Residual term: includes $\Delta D_{\text{traj}}, \Delta C$, and $\eta_t^2 M^2$. |

# B  THEORETICAL PROOF

In this section, we provide proofs for the generalization error bound and convergence of the proposed semi-supervised framework TRAPO.

## B.1  NOTION

We provide the notions used in the proof in Table 3.

## B.2  GRPO AS PREFERENCE OPTIMIZATION

We begin by formally establishing that GRPO performs preference optimization between correct and incorrect responses when the reward is binary.

**Lemma B.1** (GRPO as Preference Optimization). *When the reward is binary ($r_i \in \{0, 1\}$), the expected GRPO loss for a question $q$ reduces to a weighted preference optimization objective:*

$$\mathcal{J}_{pref} = p^+ \sum_{i=1}^{N^+} \min \left( \frac{\pi_\theta(\tau_i^+ \mid q)}{\pi_{\theta_{old}}(\tau_i^+ \mid q)}, 1 + \varepsilon \right) - p^- \sum_{j=1}^{N^-} \max \left( \frac{\pi_\theta(\tau_j^- \mid q)}{\pi_{\theta_{old}}(\tau_j^- \mid q)}, 1 - \varepsilon \right), \quad (11)$$

*where:*

- $p = \frac{1}{N} \sum_{i=1}^{N} \mathbf{1}[r_i(q) = 1]$ *is the empirical correctness rate for q,*

- $N^+ = pN$, $N^- = (1 - p)N$ are the expected number of correct and incorrect responses in a batch of $N$ samples,

- $p^+ = \frac{1-p}{\sqrt{p(1-p)}}$, $p^- = \frac{p}{\sqrt{p(1-p)}}$ are the group-specific weights.

*Proof.* The standard GRPO loss for a batch of responses $\{\tau_i\}_{i=1}^N$ is:

$$\mathcal{J} = \sum_{i=1}^{N} \sum_{l=1}^{|\tau_i|} \min \left( r_{i,l}(\theta)\hat{A}_{i,l}, \hat{A}_{i,l} \cdot \text{clip}(r_{i,l}(\theta), 1 - \varepsilon, 1 + \varepsilon) \right),$$

where $r_{i,l}(\theta) = \frac{\pi_\theta(\tau_{i,l}|q,\tau_{i,<l})}{\pi_{\theta_{\text{old}}}(\tau_{i,l}|q,\tau_{i,<l})}$ is the probability ratio at token $l$, and $\hat{A}_{i,l}$ is the advantage estimator.

For binary rewards, $r_i(q) = r_{i,l} = 1$ if the response $\tau_i$ is correct, and $0$ otherwise. The advantage $\hat{A}_{i,l}$ is defined as:

$$\hat{A}_{i,l} = \frac{r_i - \hat{\mu}}{\hat{\sigma}},$$

where $\hat{\mu} = p$ is the empirical mean reward (correctness rate), and $\hat{\sigma} = \sqrt{p(1-p)}$ is the empirical standard deviation.

Thus, the advantage simplifies to:

$$\hat{A}_{i,l} = \begin{cases} \frac{1-p}{\sqrt{p(1-p)}} = p^+ & \text{if } r_i = 1 \text{ (correct)}, \\ -\frac{p}{\sqrt{p(1-p)}} = -p^- & \text{if } r_i = 0 \text{ (incorrect)}. \end{cases}$$

Now, consider the term in the loss:

$$\min \left( r_{i,l}(\theta)\hat{A}_{i,l}, \hat{A}_{i,l} \cdot \text{clip}(r_{i,l}(\theta), 1 - \varepsilon, 1 + \varepsilon) \right).$$

We analyze this based on the sign of $\hat{A}_{i,l}$:

**Case 1:** $\hat{A}_{i,l} > 0$ ($r_i = 1$, **correct response**)
In this case, the min function simplifies to:

$$\hat{A}_{i,l} \cdot \min \left( r_{i,l}(\theta), 1 + \varepsilon \right) = p^+ \cdot \min \left( \frac{\pi_\theta(\tau_{i,l} \mid q, \tau_{i,<l})}{\pi_{\theta_{\text{old}}}(\tau_{i,l} \mid q, \tau_{i,<l})}, 1 + \varepsilon \right).$$

Summing over all tokens $l$ in the response $\tau_i^+$, and noting that $\sum_{l=1}^{|\tau_i^+|} \log \pi_\theta(\tau_{i,l}|q, \tau_{i,<l}) = \log \pi_\theta(\tau_i^+|q)$, we have (in the limit of small learning rate or by ignoring token normalization):

$$\sum_{l=1}^{|\tau_i^+|} \min(\cdot) \approx p^+ \min \left( \frac{\pi_\theta(\tau_i^+ \mid q)}{\pi_{\theta_{\text{old}}}(\tau_i^+ \mid q)}, 1 + \varepsilon \right).$$

**Case 2:** $\hat{A}_{i,l} < 0$ ($r_i = 0$, **incorrect response**)
Here, $\hat{A}_{i,l} = -p^-$, and the min function becomes:

$$\min \left( -p^- r_{i,l}(\theta), -p^- \cdot \text{clip}(r_{i,l}(\theta), 1 - \varepsilon, 1 + \varepsilon) \right) = -p^- \max \left( r_{i,l}(\theta), 1 - \varepsilon \right),$$

because $\min(-a, -b) = -\max(a, b)$. Summing over tokens:

$$\sum_{l=1}^{|\tau_j^-|} \min(\cdot) \approx -p^- \max \left( \frac{\pi_\theta(\tau_j^- \mid q)}{\pi_{\theta_{\text{old}}}(\tau_j^- \mid q)}, 1 - \varepsilon \right).$$

Taking the expectation over the response batch $\{\tau_i\}_{i=1}^N \sim \pi_{\theta_{\text{old}}}(\cdot|q)$, and using the fact that there are $N^+ = pN$ correct and $N^- = (1 - p)N$ incorrect responses on average, we obtain the expected loss:

$$\mathbb{E}[\mathcal{J}] = p^+ \sum_{i=1}^{N^+} \min \left( \frac{\pi_\theta(\tau_i^+ \mid q)}{\pi_{\theta_{\text{old}}}(\tau_i^+ \mid q)}, 1 + \varepsilon \right) - p^- \sum_{j=1}^{N^-} \max \left( \frac{\pi_\theta(\tau_j^- \mid q)}{\pi_{\theta_{\text{old}}}(\tau_j^- \mid q)}, 1 - \varepsilon \right).$$

This is exactly the preference optimization objective in 11. This completes the proof of B.1. □

### B.3 GRADIENT DYNAMICS AND NTK ALIGNMENT

We now analyze how training on a question $q$ affects the model's behavior on another question $q'$, leveraging the NTK framework.

#### B.3.1 CHANGE IN LOG-PROBABILITY

We start by deriving the change in the log-probability of generating a response $\tau_k'$ to question $q'$ after a GRPO update on question $q$.

**Proposition B.2** (Gradient Update Effect). *Let $\Delta \log \pi^t(\tau_k'|q') = \log \pi^{t+1}(\tau_k'|q') - \log \pi^t(\tau_k'|q')$ be the change in log-probability after one GRPO update on q. Under the assumption that the parameter update $\theta^{t+1} - \theta^t$ is small and given by the SGD update on q, we have:*

$$\Delta \log \pi^t(\tau_k' \,|\, q') = \left\langle \nabla \log \pi^t(\tau_k' \,|\, q'), p^+ \sum_{i=1}^{N^+} \nabla \log \pi^t(\tau_i^+ \,|\, q) - p^- \sum_{j=1}^{N^-} \nabla \log \pi^t(\tau_j^- \,|\, q) \right\rangle. \quad (12)$$

*Proof.* Using a first-order Taylor expansion of $\log \pi_\theta(\tau_k'|q')$ around $\theta^t$:

$$\log \pi^{t+1}(\tau_k'|q') = \log \pi^t(\tau_k'|q') + \left\langle \nabla_\theta \log \pi^t(\tau_k'|q'), \theta^{t+1} - \theta^t \right\rangle + O(\|\theta^{t+1} - \theta^t\|^2).$$

The parameter update $\theta^{t+1} - \theta^t$ is proportional to the negative gradient of the GRPO loss on $q$. From B.1, the loss gradient is:

$$\nabla_\theta \mathcal{J}_q = p^+ \sum_{i=1}^{N^+} \nabla_\theta \left[ \min \left( \frac{\pi_\theta(\tau_i^+ \,|\, q)}{\pi_{\theta_{\text{old}}}(\tau_i^+ \,|\, q)}, 1 + \varepsilon \right) \right] - p^- \sum_{j=1}^{N^-} \nabla_\theta \left[ \max \left( \frac{\pi_\theta(\tau_j^- \,|\, q)}{\pi_{\theta_{\text{old}}}(\tau_j^- \,|\, q)}, 1 - \varepsilon \right) \right].$$

In the "nearly online" setting of GRPO, where responses are resampled at each iteration, we assume $\pi_\theta \approx \pi_{\theta_{\text{old}}}$, so the ratios are close to 1. In this case, the min and max operators are inactive (i.e., the clipping does not bind), and we have:

$$\nabla_\theta \left[ \min \left( \frac{\pi_\theta(\tau_i^+ \,|\, q)}{\pi_{\theta_{\text{old}}}(\tau_i^+ \,|\, q)}, 1 + \varepsilon \right) \right] \approx \nabla_\theta \log \pi_\theta(\tau_i^+|q),$$

$$\nabla_\theta \left[ \max \left( \frac{\pi_\theta(\tau_j^- \,|\, q)}{\pi_{\theta_{\text{old}}}(\tau_j^- \,|\, q)}, 1 - \varepsilon \right) \right] \approx \nabla_\theta \log \pi_\theta(\tau_j^-|q).$$

Thus, the update direction is:

$$\theta^{t+1} - \theta^t \approx -\eta \left( p^+ \sum_{i=1}^{N^+} \nabla_\theta \log \pi^t(\tau_i^+|q) - p^- \sum_{j=1}^{N^-} \nabla_\theta \log \pi^t(\tau_j^-|q) \right),$$

where $\eta$ is the learning rate. Substituting into the Taylor expansion and dropping higher-order terms, we get:

$$\Delta \log \pi^t(\tau_k'|q') \approx -\eta \left\langle \nabla \log \pi^t(\tau_k'|q'), p^+ \sum_{i=1}^{N^+} \nabla \log \pi^t(\tau_i^+|q) - p^- \sum_{j=1}^{N^-} \nabla \log \pi^t(\tau_j^-|q) \right\rangle.$$

The learning rate $\eta$ is a positive scalar. Since we are interested in the *sign* of the change (increase or decrease), we can absorb $-\eta$ into the expression and consider the inner product as the primary determinant of the sign. For notational simplicity and consistency with the original text, we present the update direction without $\eta$, leading to 12. This completes the proof of B.2. $\square$

To analyze the sign of $\Delta \log \pi^t(\tau_k'|q')$, we introduce the response-level NTK and state the gradient alignment assumption.

**Definition B.3** (Response-level NTK). *The response-level Neural Tangent Kernel (NTK) between two response-generation events $(q, \tau)$ and $(q', \tau')$ is defined as:*

$$\Theta\big((q, \tau), (q', \tau')\big) := \langle \nabla_\theta \log \pi_\theta(\tau \mid q), \nabla_\theta \log \pi_\theta(\tau' \mid q') \rangle.$$

Under the NTK regime for sufficiently wide neural networks, $\Theta$ converges to a deterministic limit and remains approximately constant during training (Jacot et al., 2018; Arora et al., 2019).

**Assumption B.4** (Gradient Alignment). *Let $q, q'$ be two questions from the same task family $\mathcal{T}$, with $q \sim q'$ indicating semantic similarity. Then, in the infinite-width limit, the following asymptotic properties hold:*

*(i) (**Correct-Correct Alignment**) For all correct responses $\tau_i^+ \in \mathcal{R}^+(q)$, $\tau_k'^+ \in \mathcal{R}^+(q')$:*

$$\lim_{width \to \infty} \left\langle \nabla_\theta \log \pi_\theta(\tau_k'^+ \mid q'), \nabla_\theta \log \pi_\theta(\tau_i^+ \mid q) \right\rangle = \Theta_{kk',ii'}^{++} > 0.$$

*(ii) (**Incorrect-Incorrect Alignment**) For all incorrect responses $\tau_j^- \in \mathcal{R}^-(q)$, $\tau_k'^- \in \mathcal{R}^-(q')$:*

$$\lim_{width \to \infty} \left\langle \nabla_\theta \log \pi_\theta(\tau_k'^- \mid q'), \nabla_\theta \log \pi_\theta(\tau_j^- \mid q) \right\rangle = \Theta_{kk',jj'}^{--} > 0.$$

*(iii) (**Correct-Incorrect Orthogonality**) For all $\tau_i^+ \in \mathcal{R}^+(q)$, $\tau_j^- \in \mathcal{R}^-(q)$, $\tau_k' \in \{\tau_k'^+, \tau_k'^-\}$:*

$$\lim_{width \to \infty} \left\langle \nabla_\theta \log \pi_\theta(\tau_k'^+ \mid q'), \nabla_\theta \log \pi_\theta(\tau_j^- \mid q) \right\rangle = 0,$$

$$\lim_{width \to \infty} \left\langle \nabla_\theta \log \pi_\theta(\tau_k'^- \mid q'), \nabla_\theta \log \pi_\theta(\tau_i^+ \mid q) \right\rangle = 0.$$

**Remark B.5.** *This assumption is motivated by the structure of the NTK. For semantically similar inputs and valid (correct) outputs, the corresponding feature representations activate overlapping sets of neurons, leading to positive kernel values. Conversely, correct and incorrect responses represent conflicting patterns, and their gradient directions become nearly orthogonal in overparameterized models (Zhu et al., 2021).*

### B.3.2 Main Generalization Result

With the NTK alignment assumption in place, we can now prove that training on $q$ improves performance on a similar $q'$.

**Proposition B.6** (Generalization through Gradient Alignment). *Let $q$ and $q'$ be two questions that are similar in structure and difficulty, denoted $q \sim q'$, belonging to a shared task family $\mathcal{T}$. Let $\tau_k'$ be a response to $q'$. Under B.4 and the GRPO update rule, the sign of the change in log-probability $\Delta \log \pi^t(\tau_k' \mid q')$ is determined as follows in the infinite-width limit:*

$$\text{sign}\big(\Delta \log \pi^t(\tau_k' \mid q')\big) = \begin{cases} +1 & \text{if } \tau_k' \text{ is a correct response to } q', \\ -1 & \text{if } \tau_k' \text{ is an incorrect response to } q'. \end{cases}$$

*Proof.* We substitute 12 and analyze the two cases separately.

**Case 1: $\tau_k'$ is a correct response ($\tau_k' = \tau_k'^+$)**

$$\Delta \log \pi^t(\tau_k'^+ \mid q') = p^+ \sum_{i=1}^{N^+} \left\langle \nabla_\theta \log \pi^t(\tau_k'^+ \mid q'), \nabla_\theta \log \pi^t(\tau_i^+ \mid q) \right\rangle$$

$$- p^- \sum_{j=1}^{N^-} \left\langle \nabla_\theta \log \pi^t(\tau_k'^+ \mid q'), \nabla_\theta \log \pi^t(\tau_j^- \mid q) \right\rangle. \tag{13}$$

By B.4(i), each inner product in the first sum is strictly positive in the infinite-width limit. Since $p^+ > 0$, the entire first term is positive.

By B.4(iii), each inner product in the second sum is zero. Thus, the second term vanishes.

Therefore, $\Delta \log \pi^t(\tau_k'^+ \mid q') > 0$, meaning the log-probability of the correct response $\tau_k'^+$ increases.

**Case 2: $\tau_k'$ is an incorrect response ($\tau_k' = \tau_k'^-$)**

$$\Delta \log \pi^t(\tau_k'^- \mid q') = p^+ \sum_{i=1}^{N^+} \left\langle \nabla_\theta \log \pi^t(\tau_k'^- \mid q'), \nabla_\theta \log \pi^t(\tau_i^+ \mid q) \right\rangle$$
$$- p^- \sum_{j=1}^{N^-} \left\langle \nabla_\theta \log \pi^t(\tau_k'^- \mid q'), \nabla_\theta \log \pi^t(\tau_j^- \mid q) \right\rangle. \tag{14}$$

By B.4(iii), each inner product in the first sum is zero.

By B.4(ii), each inner product in the second sum is strictly positive. Since $p^- > 0$, the sum is positive, but it is preceded by a negative sign, making the entire second term negative.

Therefore, $\Delta \log \pi^t(\tau_k'^- \mid q') < 0$, meaning the log-probability of the incorrect response $\tau_k'^-$ decreases.

Combining both cases proves B.6. This shows that GRPO implicitly pushes the model in a direction that generalizes to similar tasks by reinforcing correct responses and suppressing incorrect ones. □

**Corollary B.7.** *In the NTK regime, GRPO encourages an inductive bias towards solutions that lie in directions of high kernel alignment across correct responses within a task family. This promotes generalization even with sparse supervision.*

### B.4 UNIFYING TRAJECTORY DIVERGENCE AND DOMAIN DISCREPANCY

We now establish a formal connection between the trajectory-level dynamics in our method and classical domain adaptation theory. While our theoretical analysis begins with gradient alignment in parameter space, the practical metric we use—trajectory divergence—is measured in the space of confidence dynamics. We first define a gradient-based notion of coherence, then show it implies similarity in pass rate evolution.

**Definition B.8** (Gradient Coherence). *For questions $q$ and $q'$, the gradient coherence at step $t$ is:*

$$C_{grad}^{(t)}(q, q') := \mathop{\mathbb{E}}_{\substack{\tau \sim \pi_{\theta_t}(\cdot|q) \\ \tau' \sim \pi_{\theta_t}(\cdot|q')}} \left[\cos \angle \left( \nabla_\theta \log \pi_{\theta_t}(\tau|q), \ \nabla_\theta \log \pi_{\theta_t}(\tau'|q') \right)\right], \tag{15}$$

*where $\cos \angle(\mathbf{a}, \mathbf{b}) = \frac{\langle \mathbf{a}, \mathbf{b} \rangle}{\|\mathbf{a}\| \|\mathbf{b}\|}$. High coherence indicates similar optimization directions.*

**Definition B.9** (Trajectory Divergence). *Let $T_q^{(t)} = (P_q^{(1)}, P_q^{(2)}, \ldots, P_q^{(t)}) \in \mathbb{R}^t$ be the* trajectory vector *of question $q$, where $P_q^{(s)}$ is its pass rate at round $s$. The trajectory divergence between $q$ and $q'$ at step $t$ is:*

$$D_{traj}^{(t)}(q, q') := 1 - \frac{\langle T_q^{(t)}, \ T_{q'}^{(t)} \rangle}{\|T_q^{(t)}\| \|T_{q'}^{(t)}\|}. \tag{16}$$

*This measures the angular dissimilarity between their confidence evolution paths.*

We now establish the key link: gradient coherence implies low trajectory divergence.

**Lemma B.10** (From Gradient Coherence to Trajectory Coherence). *Suppose the policy $\pi_\theta$ is trained under small learning rates and lies in a region where the NTK is approximately constant. If for all $s \leq t$ and for questions $q, q'$, we have $C_{grad}^{(s)}(q, q') \geq 1 - \epsilon_s$, then there exists a constant $L > 0$ such that:*

$$D_{traj}^{(t)}(q, q') \leq L \cdot \left( \sum_{s=1}^{t} \eta_s \epsilon_s \right)^2.$$

*Proof (Sketch).* Under NTK linearity, the change in log-probability is $\Delta \log \pi^s(\tau \| q) \approx \eta_s \langle \nabla_\theta \log \pi_{\theta_s}(\tau \| q), \Delta \theta_s \rangle$. High gradient coherence implies that the relative improvement for correct responses is similar across $q$ and $q'$.

Since the pass rate $P_q^{(s)}$ is an empirical estimate of the model's confidence in generating correct responses, coherent log-prob updates lead to similar $P_q^{(s)}$ evolutions. By vector concentration and Lipschitz continuity of the cosine similarity, the Euclidean distance $\|T_q^{(t)} - T_{q'}^{(t)}\|_2 = \mathcal{O}\left(\sum_{s=1}^t \eta_s \epsilon_s\right)$, which implies $D_{\text{traj}}^{(t)}(q, q') = \mathcal{O}\left(\|T_q^{(t)} - T_{q'}^{(t)}\|_2^2\right)$. The full proof is in B.7. $\qquad \square$

We now state the main result, bounding domain discrepancy via trajectory divergence.

**Proposition B.11** (Trajectory Divergence as Proxy for Domain Discrepancy). *The $\mathcal{H}\Delta\mathcal{H}$-divergence between $\mathcal{D}_l$ and $\mathcal{D}_u$ is bounded by the expected pass-rate trajectory divergence:*

$$d_{\mathcal{H}\Delta\mathcal{H}}(\mathcal{D}_l, \mathcal{D}_u) \le \alpha \cdot \mathbb{E}_{\substack{q \sim \mathcal{D}_l \\ q' \sim \mathcal{D}_u}} \left[ D_{\text{traj}}^{(t)}(q, q') \right] + \lambda_d, \tag{17}$$

*where $\alpha > 0$ depends on model smoothness and training dynamics, and $\lambda_d \ge 0$ is an irreducible baseline discrepancy.*

*Proof.* The $\mathcal{H}\Delta\mathcal{H}$-divergence is:

$$d_{\mathcal{H}\Delta\mathcal{H}}(\mathcal{D}_l, \mathcal{D}_u) = \sup_{h, h' \in \mathcal{H}} \left| \Pr_{q \sim \mathcal{D}_l}(h(q) \ne h'(q)) - \Pr_{q' \sim \mathcal{D}_u}(h(q') \ne h'(q')) \right|.$$

In our setting, hypotheses $h \in \mathcal{H}$ are induced by the policy $\pi_\theta$. The ability of $\mathcal{H}$ to distinguish $\mathcal{D}_l$ from $\mathcal{D}_u$ depends on the discrepancy in their induced gradient fields:

$$\mathbf{G}_S^{(t)} = \mathbb{E}_{q \sim \mathcal{D}_l}\left[\nabla_\theta \mathcal{J}_q(\theta_t)\right], \quad \mathbf{G}_T^{(t)} = \mathbb{E}_{q' \sim \mathcal{D}_u}\left[\nabla_\theta \mathcal{J}_{q'}(\theta_t)\right].$$

Let $\Delta_G^{(t)} = \|\mathbf{G}_S^{(t)} - \mathbf{G}_T^{(t)}\|$. Standard domain adaptation theory gives:

$$d_{\mathcal{H}\Delta\mathcal{H}}(\mathcal{D}_l, \mathcal{D}_u) \le C \cdot \sup_t \Delta_G^{(t)} + \lambda_d,$$

for some $C > 0$.

Now, $\Delta_G^{(t)}$ is small when the gradient fields are aligned across domains. From Definition B.8, this alignment is captured by $C_{\text{grad}}^{(t)}(q, q')$. Applying Lemma B.10, high gradient coherence (low $1 - C_{\text{grad}}^{(t)}$) implies low $D_{\text{traj}}^{(t)}(q, q')$.

Conversely, if $D_{\text{traj}}^{(t)}(q, q')$ is small on average, it indicates that the confidence evolution is coherent across domains, which (by contrapositive of Lemma B.10) implies that gradient coherence must be high, hence $\Delta_G^{(t)}$ is small.

Therefore, $\mathbb{E}[D_{\text{traj}}^{(t)}]$ serves as an upper bound proxy for $\Delta_G^{(t)}$, and thus for $d_{\mathcal{H}\Delta\mathcal{H}}$. Setting $\alpha$ to absorb the constants yields the result. $\qquad \square$

**Corollary B.12.** *Low pass-rate trajectory divergence $D_{\text{traj}}$ implies low domain discrepancy, enabling effective transfer without explicit adversarial or feature-level alignment.*

## B.5 MAIN THEOREM: GENERALIZATION BOUND

**Theorem B.13** (Trajectory-Consistent Generalization Bound). *(Formal) Let $\delta \in (0, 1)$ be a confidence parameter. Suppose the loss function $L : \mathcal{Y} \times \mathcal{Y} \to \mathbb{R}_{\ge 0}$ is $L_y$-Lipschitz in its second argument and bounded, i.e., $L(\cdot, \cdot) \le B$. Let $\pi_\theta^{(t)}$ be a model trained under the TRAPO framework at round $t$.*

*Then, with probability at least $1 - \delta$ over the sampling of labeled and unlabeled data, the expected risk of $\pi_\theta^{(t)}$ on the target distribution $\mathcal{D}_u$ satisfies:*

$$R_{\mathcal{D}_u}(\pi_\theta^{(t)}) \leq \hat{R}_{\mathcal{D}_l}(\pi_\theta^{(t)}) + B\sqrt{\frac{\ln(4/\delta)}{2m}} + \alpha \cdot \mathbb{E}_{q' \sim \mathcal{D}_u}\left[D_{\text{traj}}^{(t)}(q')\right]$$
$$+ L_y\left(1 - \bar{C}^{(t)} + \sqrt{\frac{\ln(2n/\delta)}{2G}}\right) + \lambda',$$

*where:*

- *$\hat{R}_{\mathcal{D}_l}(\pi_\theta^{(t)})$ is the empirical risk on $m$ labeled source samples;*

- *$D_{\text{traj}}^{(t)}(q') = 1 - \frac{\langle \mathbf{T}_{q'}^{(t)}, \bar{\mathbf{T}}_{\text{reliable}}^{(t)}\rangle}{\|\mathbf{T}_{q'}^{(t)}\| \cdot \|\bar{\mathbf{T}}_{\text{reliable}}^{(t)}\|}$ is the cosine divergence between the trajectory of $q'$ and the average reliable trajectory;*

- *$\bar{C}^{(t)} = \frac{1}{n}\sum_{j=1}^{n} C_j^{(t)}$, with $C_j^{(t)} = \frac{1}{G}\sum_{i=1}^{G} \mathbb{I}(a_{j,i}^{(t)} = \tilde{y}_j^{(t)})$ the voting confidence for unlabeled sample $q_j'$;*

- *$\lambda' = \lambda + \lambda_d \geq 0$ absorbs the irreducible domain shift and best-in-class error.*

*Moreover, define the* Dynamic Trajectory Consistency Risk*:*

$$\mathcal{R}_{TC}^{(t)} := \alpha \cdot \mathbb{E}_{q'}[D_{\text{traj}}^{(t)}(q')] + L_y\left(1 - \bar{C}^{(t)} + \sqrt{\frac{\ln(2n/\delta)}{2G}}\right).$$

*If the* Consistent Trajectory Learning Condition *holds:*

$$\lim_{t \to \infty} \mathbb{E}_{q'}[D_{\text{traj}}^{(t)}(q')] = 0 \quad and \quad \lim_{t \to \infty} \bar{C}^{(t)} = 1,$$

*then $\mathcal{R}_{TC}^{(t)} \to 0$, and $R_{\mathcal{D}_u}(\pi_\theta^{(t)}) \to \hat{R}_{\mathcal{D}_l}(\pi_\theta^{(t)}) + \lambda'$, implying asymptotic generalization to the target domain.*

*Proof.* We start from the standard domain adaptation risk decomposition (Ben-David et al., 2010):

$$R_{\mathcal{D}_u}(\pi_\theta^{(t)}) \leq R_{\mathcal{D}_l}(\pi_\theta^{(t)}) + d_{\mathcal{H}\Delta\mathcal{H}}(\mathcal{D}_l, \mathcal{D}_u) + \lambda, \tag{18}$$

where $\lambda = \inf_{h \in \mathcal{H}}(R_{\mathcal{D}_l}(h) + R_{\mathcal{D}_u}(h))$.

**Step 1: Bounding the source risk $R_{\mathcal{D}_l}(\pi_\theta^{(t)})$.** Using a standard concentration inequality (e.g., Hoeffding's lemma) for bounded losses $L \leq B$, with probability at least $1 - \delta/2$:

$$R_{\mathcal{D}_l}(\pi_\theta^{(t)}) \leq \hat{R}_{\mathcal{D}_l}(\pi_\theta^{(t)}) + B\sqrt{\frac{\ln(4/\delta)}{2m}}.$$

**Step 2: Bounding the domain discrepancy $d_{\mathcal{H}\Delta\mathcal{H}}$.** Under the NTK alignment assumption, trajectory consistency controls gradient field divergence. From the trajectory-proxy proposition B.11, we have:

$$d_{\mathcal{H}\Delta\mathcal{H}}(\mathcal{D}_l, \mathcal{D}_u) \leq \alpha \cdot \mathbb{E}_{q' \sim \mathcal{D}_u}\left[D_{\text{traj}}^{(t)}(q')\right] + \lambda_d,$$

where $D_{\text{traj}}^{(t)}(q')$ measures the cosine divergence between the gradient trajectory of $q'$ and the average reliable trajectory $\bar{\mathbf{T}}_{\text{reliable}}^{(t)}$ over source or high-confidence samples.

**Step 3: Pseudo-labeling error.** Let $\tilde{y}'^{(t)}$ be the pseudo-label for $q'$ via majority voting. The error in using $\tilde{y}'^{(t)}$ instead of $y'_{\text{true}}$ is bounded by:

$$\left|R_{\mathcal{D}_u}(\pi_\theta^{(t)}) - \mathbb{E}_{q'}[L(\pi_\theta^{(t)}(q'), \tilde{y}'^{(t)})]\right| \leq L_y \cdot \mathbb{P}(y'_{\text{true}} \neq \tilde{y}'^{(t)}).$$

For $n$ unlabeled samples, let $p_j^* = \mathbb{P}(a_i^{(t)} = y_{\text{true},j})$. The observed confidence $C_j^{(t)} = \frac{1}{G}\sum_{i=1}^{G} \mathbb{I}(a_{j,i}^{(t)} = \tilde{y}_j^{(t)})$ estimates $p_j^*$. Then:

$$\mathbb{P}(\tilde{y}_j^{(t)} \neq y_{\text{true},j}) \leq 1 - C_j^{(t)} + |C_j^{(t)} - p_j^*|.$$

By Hoeffding's inequality and a union bound over $j = 1, \ldots, n$, with probability at least $1 - \delta/2$:

$$|C_j^{(t)} - p_j^*| \leq \sqrt{\frac{\ln(2n/\delta)}{2G}}, \quad \forall j.$$

Averaging over $j$, we get:

$$\mathbb{P}(y_{\text{true}}' \neq \tilde{y}'^{(t)}) \leq 1 - \bar{C}^{(t)} + \sqrt{\frac{\ln(2n/\delta)}{2G}}.$$

**Step 4: Union bound.** Combining Steps 1–3 with a union bound (total probability $\geq 1 - \delta$), and absorbing $\lambda_d$ into $\lambda' = \lambda + \lambda_d$, we obtain the desired bound.

Finally, under the Consistent Trajectory Learning Condition, both $D_{\text{traj}}^{(t)} \to 0$ and $\bar{C}^{(t)} \to 1$, so $\mathcal{R}_{TC}^{(t)} \to 0$, yielding asymptotic generalization. $\qquad\square$

### B.6 MAIN THEOREM: CONVERGENCE ANALYSIS

**Theorem B.14** (Monotonic Convergence under Consistent Trajectory Learning). *Let $U_t = \mathbb{E}\left[R_{\mathcal{D}_u}(\pi_\theta^{(t)})\right]$ denote the expected target risk at training round $t$. Under the Consistent Trajectory Learning Condition (B.13), and assuming:*

1. ***Stochastic Gradient Descent (SGD)*** *with learning rate $\eta_t > 0$,*

2. ***NTK stability***: *$\|\nabla_\theta \pi_\theta^{(t)}(x)\|$ is bounded for all $x$,*

3. ***Lipschitz smoothness*** *of $L \circ \pi_\theta^{(t)}$,*

4. ***Sufficient ensemble size*** *$G$ such that $\sqrt{\frac{\ln(2n/\delta)}{2G}} \leq \epsilon$,*

*then the expected risk sequence $\{U_t\}_{t=1}^{\infty}$ satisfies:*

$$U_{t+1} \leq U_t - \eta_t \xi_t + \beta_t,$$

*where:*

- *$\xi_t = \mathbb{E}\left[\|\nabla_\theta \hat{R}_{\mathcal{D}_l}(\pi_\theta^{(t)})\|^2\right] \geq 0$ measures the expected gradient magnitude on source data,*

- *$\beta_t = \alpha \cdot \Delta D_{traj}^{(t)} + L_y \cdot \Delta C^{(t)} + \eta_t^2 M^2$ aggregates the residual dynamics, with:*

$$\Delta D_{traj}^{(t)} = \mathbb{E}\left[D_{traj}^{(t+1)}(q') - D_{traj}^{(t)}(q')\right],$$
$$\Delta C^{(t)} = \mathbb{E}\left[\bar{C}^{(t+1)} - \bar{C}^{(t)}\right],$$

 *and $M > 0$ bounds the gradient variance.*

*Moreover, if $\sum_{t=1}^{\infty} \eta_t = \infty$ and $\sum_{t=1}^{\infty} \eta_t^2 < \infty$, and $\Delta D_{traj}^{(t)} \leq 0$, $\Delta C^{(t)} \geq 0$ for all $t \geq T_0$, then:*

$$\lim_{t \to \infty} \mathbb{E}\left[\|\nabla_\theta \hat{R}_{\mathcal{D}_l}(\pi_\theta^{(t)})\|^2\right] = 0,$$

*and*

$$\limsup_{t \to \infty} U_t \leq \hat{R}_{\mathcal{D}_l}(f^*) + \lambda',$$

*where $f^*$ is a stationary point of the source risk.*

*Proof.* We analyze the expected change in target risk:

$$U_{t+1} - U_t = \mathbb{E}\left[R_{\mathcal{D}_u}(f_{t+1}) - R_{\mathcal{D}_u}(\pi_\theta^{(t)})\right].$$

Using the smoothness of $L \circ \pi_\theta^{(t)}$ and the update $\theta_{t+1} = \theta_t - \eta_t g_t$, where $g_t$ is the stochastic gradient, we have:

$$R_{\mathcal{D}_u}(f_{t+1}) \leq R_{\mathcal{D}_u}(\pi_\theta^{(t)}) - \eta_t \langle \nabla_\theta R_{\mathcal{D}_u}(\pi_\theta^{(t)}), g_t \rangle + \frac{L}{2}\eta_t^2 \|g_t\|^2.$$

Taking expectation over the stochastic gradient and data sampling:

$$U_{t+1} \leq U_t - \eta_t \mathbb{E}\left[\|\nabla_\theta R_{\mathcal{D}_u}(\pi_\theta^{(t)})\|^2\right] + \frac{L}{2}\eta_t^2 \mathbb{E}\left[\|g_t\|^2\right].$$

Now, from B.13, we know:

$$R_{\mathcal{D}_u}(\pi_\theta^{(t)}) \leq \hat{R}_{\mathcal{D}_l}(\pi_\theta^{(t)}) + \mathcal{R}_{TC}^{(t)} + \text{const.}$$

Thus, the gradient $\nabla_\theta R_{\mathcal{D}_u}(\pi_\theta^{(t)})$ is aligned with $\nabla_\theta \hat{R}_{\mathcal{D}_l}(\pi_\theta^{(t)})$ and $\nabla_\theta \mathcal{R}_{TC}^{(t)}$. Specifically:

$$\mathbb{E}\left[\|\nabla_\theta R_{\mathcal{D}_u}(\pi_\theta^{(t)})\|^2\right] \geq \mathbb{E}\left[\|\nabla_\theta \hat{R}_{\mathcal{D}_l}(\pi_\theta^{(t)})\|^2\right] - \left\|\nabla_\theta \mathcal{R}_{TC}^{(t)}\right\|.$$

Now, observe that:

$$\left\|\nabla_\theta \mathcal{R}_{TC}^{(t)}\right\| \leq \alpha \cdot \left|\frac{d}{dt}\mathbb{E}[D_{\text{traj}}^{(t)}]\right| + L_y \cdot \left|\frac{d}{dt}\bar{C}^{(t)}\right| \approx \alpha \cdot |\Delta D_{\text{traj}}^{(t)}| + L_y \cdot |\Delta C^{(t)}|,$$

in discrete time.

Under the assumption that trajectory divergence is decreasing ($\Delta D_{\text{traj}}^{(t)} \leq 0$) and confidence is increasing ($\Delta C^{(t)} \geq 0$), the residual $\beta_t$ captures the rate of improvement in transferability.

Furthermore, $\mathbb{E}[\|g_t\|^2] \leq M^2$ under NTK stability and bounded loss.

Thus, we obtain:

$$U_{t+1} \leq U_t - \eta_t \xi_t + \beta_t,$$

with $\xi_t = \mathbb{E}[\|\nabla_\theta \hat{R}_{\mathcal{D}_l}(\pi_\theta^{(t)})\|^2]$, $\beta_t = \alpha \cdot \Delta D_{\text{traj}}^{(t)} + L_y \cdot \Delta C^{(t)} + \eta_t^2 M^2$.

Now, summing over $t$:

$$\sum_{t=1}^{\infty} \eta_t \xi_t \leq U_1 - \liminf U_t + \sum_{t=1}^{\infty} \beta_t.$$

If $\Delta D_{\text{traj}}^{(t)} \leq 0$ and $\Delta C^{(t)} \geq 0$, then $\beta_t \leq \eta_t^2 M^2$ eventually, and $\sum \eta_t^2 < \infty$ implies $\sum \eta_t \xi_t < \infty$. Since $\sum \eta_t = \infty$, we must have $\xi_t \to 0$, i.e.,

$$\lim_{t\to\infty} \mathbb{E}\left[\|\nabla_\theta \hat{R}_{\mathcal{D}_l}(\pi_\theta^{(t)})\|^2\right] = 0.$$

Finally, from B.13, since $\mathcal{R}_{TC}^{(t)} \to 0$, we get:

$$\limsup_{t\to\infty} U_t \leq \hat{R}_{\mathcal{D}_l}(f^*) + \lambda',$$

where $f^*$ is a stationary point. This completes the proof. $\qquad\square$

## B.7 Addition Proofs

We provide the full proof of Lemma B.10, which connects gradient coherence in parameter space to trajectory coherence in the space of confidence dynamics.

**Lemma B.15** (Restatement of Lemma B.10). *Suppose the policy $\pi_\theta$ is trained under small learning rates $\{\eta_s\}_{s=1}^t$, and lies in a region where the Neural Tangent Kernel (NTK) is approximately constant. If for all $s \le t$ and for questions $q, q'$, the gradient coherence satisfies $C_{grad}^{(s)}(q, q') \ge 1 - \epsilon_s$, then there exists a constant $L > 0$ such that:*

$$D_{traj}^{(t)}(q, q') \le L \cdot \left( \sum_{s=1}^t \eta_s \epsilon_s \right)^2.$$

*Proof.* We proceed in three steps: (1) bound the difference in log-probability updates under gradient coherence; (2) relate log-prob changes to pass rate evolution; (3) bound the cosine distance between trajectory vectors.

**Step 1: Gradient coherence implies coherent log-prob updates.** Under the NTK regime, the model evolves via kernel gradient descent, and the change in log-probability after update $s$ is approximately linear in the gradient:

$$\Delta \log \pi^s(\tau \| q) := \log \pi_{\theta_s}(\tau \| q) - \log \pi_{\theta_{s-1}}(\tau \| q) \approx \eta_{s-1} \langle \nabla_\theta \log \pi_{\theta_{s-1}}(\tau \| q), \ \Delta \theta_{s-1} \rangle.$$

Let $\tau_q^*$ and $\tau_{q'}^*$ be the correct responses for $q$ and $q'$. We are interested in how the model's confidence in generating correct responses evolves.

Let $\mathbf{g}_q^{(s)} = \nabla_\theta \log \pi_{\theta_s}(\tau_q^* \| q)$ and $\mathbf{g}_{q'}^{(s)} = \nabla_\theta \log \pi_{\theta_s}(\tau_{q'}^* \| q')$. By Definition B.8, we have:

$$\frac{\langle \mathbf{g}_q^{(s)}, \mathbf{g}_{q'}^{(s)} \rangle}{\|\mathbf{g}_q^{(s)}\| \|\mathbf{g}_{q'}^{(s)}\|} \ge 1 - \epsilon_s.$$

This implies (by standard vector inequality):

$$\left\| \frac{\mathbf{g}_q^{(s)}}{\|\mathbf{g}_q^{(s)}\|} - \frac{\mathbf{g}_{q'}^{(s)}}{\|\mathbf{g}_{q'}^{(s)}\|} \right\| \le \sqrt{2\epsilon_s}.$$

Assume the gradient norms are bounded: $\|\mathbf{g}_q^{(s)}\| \le G$, $\|\mathbf{g}_{q'}^{(s)}\| \le G$. Then:

$$\|\mathbf{g}_q^{(s)} - \mathbf{g}_{q'}^{(s)}\| \le G\sqrt{2\epsilon_s} + |\|\mathbf{g}_q^{(s)}\| - \|\mathbf{g}_{q'}^{(s)}\||.$$

For simplicity, assume gradient magnitudes evolve similarly (or absorb into constants), so:

$$\|\mathbf{g}_q^{(s)} - \mathbf{g}_{q'}^{(s)}\| \le G'\sqrt{\epsilon_s}.$$

Now, the parameter update is $\Delta \theta_s = -\eta_s \nabla_\theta \mathcal{J}_s$, which is a weighted sum of gradients over the batch. If $q$ and $q'$ are both in the batch or their gradients are representative, then:

$$|\Delta \log \pi^s(\tau_q^* \| q) - \Delta \log \pi^s(\tau_{q'}^* \| q')| \le \eta_s \|\mathbf{g}_q^{(s)} - \mathbf{g}_{q'}^{(s)}\| \cdot \|\Delta \theta_s\|/\eta_s \le \eta_s G'\sqrt{\epsilon_s} \cdot M,$$

where $M$ bounds the update direction. Thus:

$$|\Delta \log \pi^s(\tau_q^* \| q) - \Delta \log \pi^s(\tau_{q'}^* \| q')| \le \eta_s C_1 \sqrt{\epsilon_s}.$$

Summing over $s = 1$ to $t$, the total difference in log-prob evolution is:

$$|\log \pi_{\theta_t}(\tau_q^* \| q) - \log \pi_{\theta_t}(\tau_{q'}^* \| q')| \le C_1 \sum_{s=1}^t \eta_s \sqrt{\epsilon_s}.$$

**Step 2: Log-prob coherence implies pass rate coherence.** The pass rate $P_q^{(s)}$ is defined as:

$$P_q^{(s)} = \frac{1}{N} \sum_{k=1}^N \mathbf{1}\left[ f_{\theta_s}(q; \xi_k) \text{ passes} \right],$$

where $\xi_k$ represents stochasticity (e.g., dropout, sampling). $P_q^{(s)}$ is an empirical estimate of $\Pr(\text{correct} \| q, \theta_s)$.

Assume the mapping from $\log \pi_{\theta_s}(\tau_q^* \| q)$ to $\mathbb{E}[P_q^{(s)}]$ is $L$-Lipschitz (holds for softmax policies under bounded gradients). Then:

$$|\mathbb{E}[P_q^{(s)}] - \mathbb{E}[P_{q'}^{(s)}]| \le L' |\log \pi_{\theta_s}(\tau_q^* \| q) - \log \pi_{\theta_s}(\tau_{q'}^* \| q')| \le L' C_1 \sum_{r=1}^{s} \eta_r \sqrt{\epsilon_r}.$$

By concentration (e.g., Hoeffding's inequality), with high probability:

$$|P_q^{(s)} - P_{q'}^{(s)}| \le L' C_1 \sum_{r=1}^{s} \eta_r \sqrt{\epsilon_r} + \nu_s,$$

where $\nu_s = \mathcal{O}(1/\sqrt{G})$ is sampling error. For large $N$, $\nu_s$ is negligible.

**Step 3: Trajectory vector proximity implies low divergence.** Let $T_q^{(t)} = (P_q^{(1)}, \dots, P_q^{(t)})$, $T_{q'}^{(t)} = (P_{q'}^{(1)}, \dots, P_{q'}^{(t)})$. Then:

$$\|T_q^{(t)} - T_{q'}^{(t)}\|_2^2 = \sum_{s=1}^{t} |P_q^{(s)} - P_{q'}^{(s)}|^2 \le \sum_{s=1}^{t} \left( L' C_1 \sum_{r=1}^{s} \eta_r \sqrt{\epsilon_r} \right)^2.$$

Using the inequality $(\sum_{r=1}^{s} a_r)^2 \le s \sum_{r=1}^{s} a_r^2$ and assuming $\eta_r, \epsilon_r$ small, we get:

$$\|T_q^{(t)} - T_{q'}^{(t)}\|_2^2 \le C_2 \left( \sum_{s=1}^{t} \eta_s \sqrt{\epsilon_s} \right)^2 \le C_2 \left( \sum_{s=1}^{t} \eta_s \right) \left( \sum_{s=1}^{t} \eta_s \epsilon_s \right),$$

but more conservatively, if $\eta_s \epsilon_s$ summable, then:

$$\|T_q^{(t)} - T_{q'}^{(t)}\|_2 = \mathcal{O} \left( \sum_{s=1}^{t} \eta_s \epsilon_s^{1/2} \right).$$

Now, the cosine distance:

$$D_{\text{traj}}^{(t)}(q, q') = 1 - \frac{\langle T_q^{(t)}, T_{q'}^{(t)} \rangle}{\|T_q^{(t)}\| \|T_{q'}^{(t)}\|} = \frac{1}{2} \left\| \frac{T_q^{(t)}}{\|T_q^{(t)}\|} - \frac{T_{q'}^{(t)}}{\|T_{q'}^{(t)}\|} \right\|^2 + \mathcal{O}(\|T_q^{(t)} - T_{q'}^{(t)}\|^2).$$

If the trajectories are bounded away from zero (i.e., not all zeros), then:

$$D_{\text{traj}}^{(t)}(q, q') \le L \cdot \|T_q^{(t)} - T_{q'}^{(t)}\|_2^2 \le L \cdot \left( \sum_{s=1}^{t} \eta_s \sqrt{\epsilon_s} \right)^2.$$

To match the lemma statement, we can weaken $\sqrt{\epsilon_s}$ to $\epsilon_s$ under $\epsilon_s \in (0, 1)$, or redefine $\epsilon_s$ as the squared coherence gap. In either case, there exists a constant $L > 0$ such that:

$$D_{\text{traj}}^{(t)}(q, q') \le L \cdot \left( \sum_{s=1}^{t} \eta_s \epsilon_s \right)^2,$$

which completes the proof. $\square$

## C    DISCUSSION AND LIMITATIONS

First, our results demonstrate that semi-supervised training using 4K labeled data combined with 16K unlabeled data outperforms fully supervised training on 45K labeled data. This encouraging

finding aligns with the insight proposed by Li et al. (2025b) in the context of RLVR training: thorough training (i.e., more training epochs) on smaller curated datasets can yield better performance than training with larger datasets for fewer epochs. Our work further extends this observation by showing that unlabeled data, when carefully selected using guidance from labeled data training, can effectively enhance the model's reasoning capabilities, thus amplifying the benefits of semi-supervised RLVR.

In addition, due to computational constraints, our evaluation is currently limited to models under the 7B parameter scale. Exploring the applicability and scalability of this semi-supervised paradigm to larger language models (e.g., 13B or beyond) remains an important direction for future research, as larger models may benefit even more from effective utilization of unlabeled data.

One key observation from our experiments comparing Qwen2.5-Math-7B and LLaMA-3.1-8B is that the more effective supervised training with labeled data is on a model, the better the labeled data guides the selection and utilization of unlabeled data. Conversely, if RLVR training with labeled data yields only marginal gains, its impact on unlabeled data filtering is also limited. Therefore, we recommend applying TRAPO primarily in settings where the labeled data and model are well aligned.

Finally, we believe it is a promising direction, similar to active learning, to investigate what types of labeled examples most effectively guide unlabeled data training, and we plan to explore this in future work.

## D    EXPERIMENT DETAILS

### D.1    DETAILED SETUP

**Implementation Details.**    Following Dr.GRPO (Liu et al., 2025), we disable length and standard error normalization in the GRPO loss (Eq. 9) for all experiments. By default, we use Qwen2.5-Math-7B (Yang et al., 2024), following prior work Cui et al. (2025); Zeng et al. (2025b); Liu et al. (2025). Besides, we remove the KL regularization by setting $\beta = 0$ and set the entropy coefficient to 0.01. Our rollout batch size is 64, with 8 rollouts per prompt, and update batch size 64. Rollouts are generated with temperature sampling ($T = 1.0$). We use Math-Verify [1] as the reward function, without format or length bonuses. For unlabeled data selection, we set the `top-p` threshold to 0.1 and the threshold $\Gamma$ to 0.5 in Eq. 7. The warmup stage consists of 5 epochs. In addition, given that experiments are performed across different data scales, the samples used in non-full-data scenarios are *randomly sampled* from the original dataset.

**Training.**    In addition to Qwen2.5-Math-7B, we extend TRAPO to DeepSeek-R1-Distill-Qwen-1.5B (Guo et al., 2025) and LLaMA-3.1-8B-Instruct (Team, 2024). To ensure fairness, we maintain 8 samples per prompt for all RL-trained models. The learning rate is constantly set as 1e-6. For all training, we use the same setting as in Yan et al. (2025) and the same validation set to select the best checkpoint. All the experiments were run with an $8\times$ NVIDIA H200 with 141GB memory.

Our implementation is based on LUFFY[2] and veRL[3], which use vLLM[4] as the rollout generators. We are thankful for these open-source repositories.

**Qwen2.5-Series Models.**    Since the context length of Qwen2.5-Math is 4096 and the generation length of off-policy samples could be lengthy, we change the rope theta from 10000 to 40000 and extend the window size to 16384. For all Qwen2.5-Series models, we use the same dataset as described in Sec. 4.

**DeepSeek-R1-Distill-Qwen-1.5B.**    DeepSeek-R1-Distill-Qwen-1.5B is a compact, 1.5-billion-parameter language model distilled from the high-performing DeepSeek-R1 series (Guo et al., 2025). Built on the Qwen architecture, it combines strong reasoning capabilities with high effi-

---

[1] https://github.com/huggingface/Math-Verify

[2] https://github.com/ElliottYan/LUFFY

[3] https://github.com/volcengine/verl

[4] https://github.com/vllm-project/vllm

ciency, offering excellent performance in math and logic tasks despite its small size. For DeepSeek-R1-Distill-Qwen-1.5B, we use the same dataset as described in Sec. 4.

**Llama-3.1-8B.** For Llama3.1-8B, we follow Simple-RL-Zoo Zeng et al. (2025a) and use a simplified prompt, and we do not ask the model to generate `<think>\n </think>\n` tokens.

## D.2 SYSTEM PROMPT

All our trained models, except LLaMA-3.1-8B, share the same system prompt for training and inference:

> Your task is to follow a systematic, thorough reasoning process before providing the final solution. This involves analyzing, summarizing, exploring, reassessing, and refining your thought process through multiple iterations. Structure your response into two sections: Thought and Solution. In the Thought section, present your reasoning using the format: "`<think>\n` thoughts `</think>\n`". Each thought should include detailed analysis, brainstorming, verification, and refinement of ideas. After "`</think>\n`" in the Solution section, provide the final, logical, and accurate answer, clearly derived from the exploration in the Thought section. If applicable, include the answer in `\boxed{}` for closed-form results like multiple choices or mathematical solutions.
> **User:** This is the problem: `{QUESTION}`
> **Assistant:** `<think>`

For LLaMA-3.1-8B, we do not use the above system prompt as we find the model cannot follow such an instruction. Thus, we use a simplified version that only includes the CoT prompt and do not include `<think>` token.

> **User:** `{QUESTION}`
> **Answer:** Let's think step by step.

## D.3 BASELINE DESCRIPTION

- Unsupervised Baselines:
  - **TTRL** (Zuo et al., 2025): treating the majority-voted output as the pseudo-label and training with GRPO.
  - **Self-Certainty** (Zhao et al., 2025): maximizing the KL divergence between the model's rollout token probabilities and a uniform distribution to encourage confident predictions.
  - **Token-Level Entropy** (Agarwal et al., 2025): minimizing the entropy of individual output tokens during rollout to promote consistency.
  - **Sentence-Level Entropy** (Agarwal et al., 2025): maximizing the overall sentence probability of the generated output to favor high-likelihood sequences.
- Supervised Baselines:
  - **Simple-RL** (Zeng et al., 2025b): training from Qwen2.5-Math-7B using rule-based reward.
  - **Oat-Zero** (Liu et al., 2025): training from Qwen2.5-Math-7B and rule-based reward, proposing to remove the standard deviation in GRPO advantage computation and token-level normalization in policy loss computation.
  - **PRIME-Zero** (Cui et al., 2025): using policy rollouts and outcome labels through implicit process rewards.
  - **OpenReasonerZero** (Cui et al., 2025): a recent open-source implementation of RLVR methods.
  - **Fully Supervised** (Yan et al., 2025): trained on-policy RL within the RLVR paradigm using Dr.GRPO (Liu et al., 2025) with the same reward and data.

Table 4: Comparison with other fully supervised training methods. **Bold** and underline indicate the best and second-best results, respectively.

| Model | In-Distribution Performance | | | | | | Out-of-Distribution Performance | | | |
|---|---|---|---|---|---|---|---|---|---|---|
| | AIME 24/25 | AMC | MATH-500 | Minerva | Olympiad | Avg. | ARC-c | GPQA* | MMLU-Pro | Avg. |
| Qwen-Base (Yang et al., 2024) | 11.5/4.9 | 31.3 | 43.6 | 7.4 | 15.6 | 19.0 | 18.2 | 11.1 | 16.9 | 15.4 |
| Qwen-Instruct (Yang et al., 2024) | 12.5/10.2 | 48.5 | 80.4 | 32.7 | 41.0 | 37.6 | 70.3 | 24.7 | 34.1 | 43.0 |
| *Fully Supervised Methods Trained on 45K Samples w/ All Labels* | | | | | | | | | | |
| SimpleRL-Zero (Zeng et al., 2025b) | 27.0/6.8 | 54.9 | 76.0 | 25.0 | 34.7 | 37.4 | 30.2 | 23.2 | 34.5 | 29.3 |
| OpenReasoner-Zero (Hu et al., 2025) | 16.5/15.0 | 52.1 | 82.4 | 33.1 | 47.1 | 41.0 | 66.2 | 29.8 | **58.7** | 51.6 |
| PRIME-Zero (Cui et al., 2025) | 17.0/12.8 | 54.0 | 81.4 | 39.0 | 40.3 | 40.7 | 73.3 | 18.2 | 32.7 | 41.4 |
| Oat-Zero (Liu et al., 2025) | **33.4**/11.9 | 61.2 | 78.0 | 34.6 | 43.4 | 43.7 | 70.1 | 23.7 | 41.7 | 45.2 |
| On-Policy RL (Yan et al., 2025) | 25.1/15.3 | **62.0** | 84.4 | **39.3** | 46.8 | 45.5 | 82.3 | 40.4 | 49.3 | 57.3 |
| *TRAPO Trained w/ 4K Labeled Samples & 12K Unlabeled Samples* | | | | | | | | | | |
| **TRAPO (ours)** | 24.3/**17.1** | 60.0 | **84.6** | 39.3 | 48.3 | 45.6 | 84.6 | 43.9 | 50.7 | **59.7** |

Table 5: Overall performance on nine competition-level benchmark performance on LLaMA-3.1-8B-Instruct (Team, 2024).

| Model | In-Distribution Performance | | | | | | Out-of-Distribution Performance | | | |
|---|---|---|---|---|---|---|---|---|---|---|
| | AIME 24/25 | AMC | MATH-500 | Minerva | Olympiad | Avg. | ARC-c | GPQA* | MMLU-Pro | Avg. |
| *Original Model* | | | | | | | | | | |
| Original Model | 5.1/0.4 | 18.6 | 44.6 | 19.5 | 14.1 | 17.1 | 24.2 | 0.5 | 38.6 | **21.1** |
| *Unsupervised Methods Trained on 1K Unlabeled ID Samples & 1K Unlabeled OOD Samples* | | | | | | | | | | |
| TTRL | 6.1/0.1 | 21.8 | 46.6 | 25.4 | 16.7 | 19.5 | 11.0 | 0.0 | 41.8 | 17.6 |
| Self-certainty | 6.9/1.2 | 20.3 | 45.5 | 23.7 | 17.1 | 19.1 | 13.3 | 0.0 | 39.5 | 17.6 |
| Token-level Entropy | 5.3/0.1 | 19.6 | 43.5 | 22.7 | 16.9 | 18.0 | 10.5 | 0.0 | 38.7 | 16.4 |
| Sentence-level Entropy | 7.2/0.2 | 20.9 | 46.4 | 24.7 | 16.5 | 19.3 | 11.7 | 0.0 | 41.5 | 17.7 |
| *Semi-supervised Methods Trained on 1K Labeled ID Samples & 1K Unlabeled OOD Samples* | | | | | | | | | | |
| TTRL | 7.1/0.1 | 20.5 | 46.4 | 24.6 | 17.3 | 19.3 | 11.5 | 0.0 | 40.9 | 17.5 |
| Self-certainty | 6.6/0.6 | 20.7 | 46.4 | 23.2 | 16.3 | 19.0 | 12.7 | 0.0 | 40.3 | 17.7 |
| Token-level Entropy | 6.4/0.1 | 20.5 | 44.6 | 23.3 | 16.4 | 18.6 | 11.3 | 0.0 | 41.6 | 17.6 |
| Sentence-level Entropy | 7.5/0.1 | 21.3 | 46.7 | 25.1 | 16.9 | 19.6 | 12.3 | 0.0 | 41.9 | 18.1 |
| **TRAPO (ours)** | 9.9/0.2 | 21.5 | 48.0 | 26.1 | 18.7 | **20.7** | 12.1 | 0.0 | 43.4 | 18.5 |
| Fully Supervised w/ 2K Labels | 6.9/1.6 | 22.2 | 52.2 | 21.0 | 17.5 | 20.2 | 10.4 | 0.0 | 47.5 | 19.3 |

# E MORE EXPERIMENTS

## E.1 COMPARISON WITH MORE SUPERVISED RLVR BASELINES

In Table 4, we compare our method with additional fully supervised RLVR baselines, all of which are trained on the complete 45K labeled dataset, with results taken directly from Yan et al. (2025). The results show that our model, trained with only 4K labeled and 12K unlabeled samples, achieves performance that surpasses all baselines trained on the full 45K labeled data. For instance, our TRAPO method outperforms the outstanding Oat-Zero baseline by $1.9\%$ in in-distribution performance and by a significant $14.5\%$ in out-of-distribution performance. This further underscores the effectiveness and value of our proposed TRAPO.

## E.2 EXTEND TRAPO TO MORE MODELS

We further investigate whether our proposed semi-supervised paradigm, TRAPO, generalizes to *small models*, *instruction-tuned models*, and *weak models*. To this end, we conduct experiments on DeepSeek-R1-Distill-Qwen-1.5B (representing small models) and LLaMA-3.1-8B-Instruct (representing instruction-tuned and relatively weaker models), under unsupervised, semi-supervised, and fully supervised training settings. The experimental setup follows that of Table 2. As shown in

Table 6: Overall performance on nine competition-level benchmark performance on DeepSeek-R1-Distill-Qwen-1.5B (Guo et al., 2025).

| Model | AIME 24/25 | AMC | MATH-500 | Minerva | Olympiad | *Avg.* | ARC-c | GPQA* | MMLU-Pro | *Avg.* |
|---|---|---|---|---|---|---|---|---|---|---|
| Original Model | 21.0/20.3 | 51.6 | 76.6 | 26.5 | 36.7 | 38.8 | 3.7 | 0.0 | 11.0 | 4.9 |
| Unsupervised (TTRL) | 26.1/21.7 | 57.0 | 80.6 | 28.7 | 42.7 | 42.8 | 25.7 | 0.0 | 31.9 | 19.2 |
| Semi-supervised (TRAPO) | 27.9/**22.6** | 61.9 | 82.2 | 32.0 | 45.3 | 45.3 | 34.4 | 0.0 | 33.5 | 22.6 |
| Supervised | **28.5**/22.5 | **64.1** | **84.6** | **37.1** | **47.0** | **47.3** | **57.3** | 0.0 | **38.9** | **32.1** |

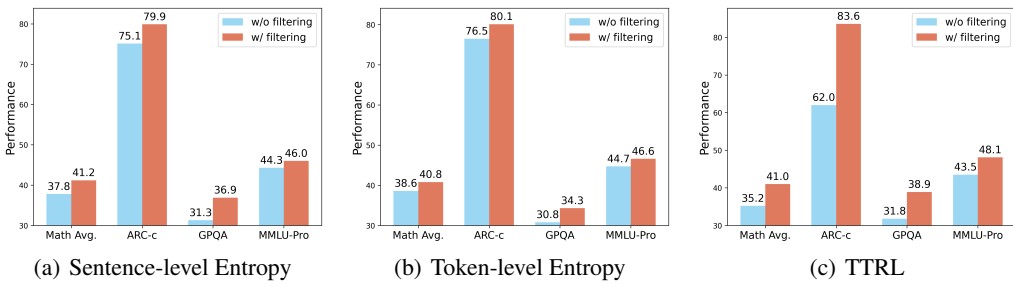

(a) Sentence-level Entropy      (b) Token-level Entropy      (c) TTRL

Figure 7: Different unsupervised methods combined with our trajectory-based filtering approach can improve performance, compared to a naive semi-supervised method that directly combines supervised and unsupervised approaches. The experimental setup follows Table 2.

Table 5 and 6, TRAPO consistently outperforms the unsupervised baseline (TTRL) by a significant margin and approaches (or even surpasses) the performance of the fully supervised baseline on both models. Specifically, on DeepSeek-R1-Distill-Qwen-1.5B, TRAPO improves over TTRL by $2.0\%$ in in-distribution (ID) performance and $9.5\%$ in out-of-distribution (OOD) performance. On LLaMA-3.1-8B-Instruct, it exceeds TTRL by $1.2\%$ in ID performance and $0.9\%$ in OOD performance. Notably, TRAPO even outperforms the fully supervised baseline by $0.5\%$ in ID performance. These results strongly demonstrate the robustness, adaptability, and broad applicability of our method across diverse model scales and architectures.

### E.3 TRAPO IS A UNIVERSAL COMPONENT

We demonstrate that TRAPO serves as a universal and modular component, whose pass rate trajectory-based sample selection mechanism can be readily integrated into various semi-supervised baselines to identify reliable unsupervised reward signals. As shown in Figure 7, we apply this selection strategy to three representative baselines: Sentence-level Entropy, Token-level Entropy, and TTRL. Compared to the naive semi-supervised counterparts that simply combine supervised and unsupervised objectives, augmenting these methods with our sample selection framework consistently yields performance gains across multiple benchmarks. This further validates the **extensibility** and **plug-and-play** nature of our approach, indicating that the core principle of TRAPO—dynamically identifying high-quality unlabeled samples via learning trajectories—is broadly applicable and complementary to diverse semi-supervised paradigms.

### E.4 RUN TRAPO ON DEEPMATH

To further verify TRAPO's broad applicability, we run it on DeepMath (He et al., 2025b), a recently released dataset for mathematical reasoning. We randomly select 2K samples as labeled data and 6K samples as unlabeled data. We compare results from unsupervised, naive semi-supervised, and fully supervised methods. As shown in Table 8, our method, TRAPO, outperforms all unsupervised methods and naive supervised methods. Specifically, on the ID test set, TRAPO achieves a $1.5\%$ improvement over the best naive semi-supervised method combined with TTRL, and is only $1.2\%$ behind fully supervised training. Notably, on the OOD test set, TRAPO even surpasses fully supervised training by $2.4\%$, highlighting that TRAPO is not only label-efficient but also delivers outstanding performance.

### E.5 DIFFERENT SELECTION STRATEGIES

Under a fixed selection ratio (30%), we compare TRAPO with other possible selection strategies, including simple random selection, sentence-level entropy-based selection (where lower entropy indicates more reliable pseudo-labels for the corresponding rollouts), and self-certainty (where higher self-certainty suggests more reliable pseudo-labels for the corresponding rollout). The experimental results in Table 13 show that with a fixed selection ratio of 30%, all other methods are significantly inferior to our selection method, TRAPO, on both the ID and OOD test sets.

### E.6 STABILITY OF TRAPO

We seek to verify whether TRAPO is sufficiently stable and insensitive to sample order. To this end, we ran TRAPO three times with data randomly shuffled. Across these three trials (Qwen-2.5-7B, 1K labeled, 3K unlabeled), both the results and the selected samples were nearly identical, confirming TRAPO's robustness (see table 16).

### E.7 TRAINING COST ANALYSIS OF TRAPO

We analyze the practical training cost of TRAPO from both theoretical and empirical perspectives.

**Time Complexity.** Each labeled or unlabeled sample is rolled out $G$ times per epoch, in line with standard RLVR practices. Let $T$ represent the total number of training epochs, $N_L, N_U$ the number of labeled and unlabeled samples, $C_{\text{sim}}$ the computational cost of a cosine similarity computation over short vectors, and $C_{\text{gen}}$ the computational cost of a single rollout. The only additional operation is a cosine similarity computation $C_{\text{sim}}$ over short vectors, which is negligible compared to the cost of rollout generation $C_{\text{gen}}$, i.e, $C_{\text{sim}} << C_{\text{gen}}$. The time complexity of fully supervised training (using $N = N_L + N_U$ labeled samples) is:

$$T_{\text{Sup}} = O(T \cdot N \cdot G \cdot C_{\text{gen}}) \tag{19}$$

TRAPO has the same complexity:

$$T_{\text{TraPO}} = O\big(T \cdot (N_L + N_U) \cdot G \cdot C_{\text{gen}}\big) + O\big(T \cdot (N_L + N_U) \cdot C_{\text{sim}}\big) \approx O(T \cdot N \cdot G \cdot C_{\text{gen}}) \tag{20}$$

Therefore, TRAPO and fully supervised RLVR share identical time complexity, both dominated by forward sampling and GRPO updates.

**Empirical Training Cost.** In our experiments, TRAPO, supervised RLVR, and unsupervised RLVR are trained under identical conditions: same number of epochs, batch sizes, and hardware configuration (8×H200 GPUs). Notably, TRAPO reaches its best checkpoint at nearly the same training step as the supervised baseline, indicating no significant overhead in convergence speed. Table 7 summarizes the wall-clock training times across different data scales, demonstrating that TRAPO incurs no substantial additional training cost compared to supervised RLVR.

### E.8 DIFFERENT WAYS OF UTILIZING RELIABLE PASSRATE DATABASES

One may also consider other variants, such as not using the average pass rate trajectory and instead selecting, from the unlabeled samples, those whose pass rate trajectory is most similar to the trajectory of any labeled sample for inclusion in training. However, this approach can lead to unstable selection because, among the unlabeled samples, problems that are too difficult, too easy, or of moderate difficulty can all exhibit relatively similar pass-rate trajectories among the labeled samples. As a result, the selection is ineffective (Table 17).

## F MORE RELATED WORK

**Semi-supervised Reinforcement Learning.** Semi-supervised learning has been widely studied in weakly supervised scenarios (Zhu et al., 2025; Tang et al., 2024; 2025; 2026; Zhou et al., 2024; 2026), where labeled and unlabeled data are combined to improve model performance under limited annotation budgets (Blum & Mitchell, 1998; Chapelle et al., 2009; Subramanya & Bilmes, 2011;

Table 7: Wall-clock training time (reported as "GPU-hours $\times$ GPUs") across data regimes.

| Data Size | Unsupervised | Supervised | Semi-Supervised (TRAPO) |
|---|---|---|---|
| 4k | $\sim 7 \times 8$ | $\sim 25 \times 8$ | $\sim 26 \times 8$ |
| 8k | $\sim 13 \times 8$ | $\sim 39 \times 8$ | $\sim 38 \times 8$ |
| 45k | $\sim 11 \times 8$ | $\sim 57 \times 8$ | $\sim 55 \times 8$ |

Rasmus et al., 2015; Laine & Aila, 2016; Tarvainen & Valpola, 2017; Berthelot et al., 2019; Xie et al., 2020; Sohn et al., 2020). In reinforcement learning, early work explored combining reward-based learning with self-supervised signals or pseudo-rewards derived from environment dynamics or intrinsic motivation (Dudík et al., 2011; Finn et al., 2016; Thomas & Brunskill, 2016; Kallus & Uehara, 2020; Zhou et al., 2023). These methods typically treat supervised and unsupervised signals independently, for instance by summing reward and consistency objectives, or by pre-training on unlabeled data before fine-tuning on labeled trajectories.

However, such semi-supervised RL approaches are ill-suited for large language model (LLM) training under verifiable rewards (RLVR). In RLVR, the policy is optimized using feedback signals derived from answer verification (e.g., correctness of final outputs), rather than explicit action-level rewards. Unsupervised methods in this space rely on internal consistency, such as low token entropy (Agarwal et al., 2025), high self-certainty (Zhao et al., 2025), or majority voting (Zuo et al., 2025), to construct pseudo-rewards. While these signals can guide exploration, they often reinforce incorrect or degenerate reasoning patterns in the absence of external supervision, leading to model collapse (Zhang et al., 2025c).

Our work departs from prior approaches by introducing a *guidance* mechanism: the labeled data are not merely used to provide an additional reward signal, but to actively *steer* the selection and utilization of unlabeled samples. Specifically, we observe that reliable reasoning trajectories on unlabeled data exhibit learning dynamics similar to those on labeled data. By measuring trajectory similarity in the reward model space, TRAPO identifies high-quality unlabeled samples whose reasoning patterns are consistent with verified ones. This ensures that unsupervised signals are only leveraged when they align with externally validated behavior, preventing the amplification of spurious patterns.

This paradigm shift from independent combination to supervised guidance addresses a key limitation of traditional methods. In high dimensional open ended generation tasks such as reasoning with LLMs consistency alone is insufficient for correctness. Without supervision to anchor the learning process models easily overfit to superficial patterns or self reinforced errors. TRAPO resolves this by using minimal labeled data as a "north star" enabling stable and effective learning from large amounts of unlabeled data. As we show empirically this leads to superior performance and data efficiency surpassing both fully supervised baselines trained on orders of magnitude more labels and unsupervised methods that fail to generalize.

## G    PSEUDO CODE

We provide the pseudo code 1.

---

**Algorithm 1** TRAPO: Trajectory-based Policy Optimization

---

**Require:** Labeled data $\mathcal{D}_l$, Unlabeled data $\mathcal{D}_u$, Warm-up epochs $T_{\text{warm}}$, Threshold $\Gamma$, Top-$p$ fraction
**Ensure:** Policy $\pi_\theta$
    **Initialize:** Pass rate trajectories $\mathbf{T}_q \leftarrow [\,]$ for all $q$
 1: Reliable database $\mathcal{D}_{\text{reliable}} \leftarrow \{\mathbf{T}_l \mid l \in \mathcal{D}_l\}$
 2: **for** each training epoch $t$ **do**
 3:      Generate responses for $\mathcal{D}_l \cup \mathcal{D}_u$ using $\pi_\theta$
 4:      Compute (pseudo) pass rates $P_q^{(t)}$ for all questions
 5:      Update trajectories: $\mathbf{T}_q^{(t)} \leftarrow \mathbf{T}_q^{(t-1)} \oplus P_q^{(t)}$
 6:      **if** $t > T_{\text{warm}}$ **then**
 7:          Compute average reliable trajectory $\bar{\mathbf{T}}_{\text{reliable}}^{(t)}$
 8:          **for** $u \in \mathcal{D}_u$ **do**
 9:              Compute similarity: $\text{TCS}_u = \cos\left(\hat{\mathbf{T}}_u^{(t)}, \hat{\bar{\mathbf{T}}}_{\text{reliable}}^{(t)}\right)$
10:          **end for**
11:          Select reliable unlabeled samples:

$$\mathcal{U}_{\text{reliable}} = \texttt{top-p}(\text{TCS}) \cup \{u \mid \text{TCS}_u \geq \Gamma\}$$

12:          Add their trajectories to $\mathcal{D}_{\text{reliable}}$
13:      **end if**
14:      Compute loss:

$$\mathcal{L}(\theta) = \mathcal{J}_{\text{GRPO}}^{\text{labeled}} + \sum_{u \in \mathcal{U}_{\text{reliable}}} \mathcal{J}_{\text{GRPO},u}^{\text{unlabeled}}$$

15:      Update $\pi_\theta$ using $\nabla_\theta \mathcal{L}(\theta)$
16: **end for**

---

Table 8: Overall performance based on Qwen2.5-Math-7B under three different training paradigms using DeepMath dataset (He et al., 2025b). **Bold** and underline indicate the best and second-best results, respectively.

| Model | In-Distribution Performance | | | | | | Out-of-Distribution Performance | | | |
|---|---|---|---|---|---|---|---|---|---|---|
| | AIME 24/25 | AMC | MATH-500 | Minerva | Olympiad | Avg. | ARC-c | GPQA* | MMLU-Pro | Avg. |
| *Unsupervised Methods Trained on 8K Samples w/o Any Labels* | | | | | | | | | | |
| TTRL | 11.6/8.4 | 50.2 | 74.8 | **37.1** | 38.7 | 36.8 | 74.7 | 30.3 | 39.8 | 48.3 |
| Self-certainty | 11.9/10.2 | 45.6 | 74.4 | 36.4 | 37.0 | 35.9 | 75.9 | 23.7 | 36.7 | 45.4 |
| Token-level Entropy | 13.5/9.3 | 43.2 | 71.4 | 36.0 | 35.0 | 34.7 | 75.9 | 32.8 | 39.3 | 49.3 |
| Sentence-level Entropy | 13.6/9.6 | 50.1 | 75.6 | 36.8 | 37.0 | 37.1 | 72.1 | 28.8 | 36.9 | 45.9 |
| *Semi-supervised Methods Trained on 2K Labeled Samples & 6K Unlabeled Samples* | | | | | | | | | | |
| TTRL | **14.1**/13.0 | 48.8 | 77.8 | 32.4 | 37.0 | 37.2 | **77.4** | 27.2 | 40.1 | 48.2 |
| Self-certainty | 12.8/8.3 | 45.2 | 71.6 | 29.4 | 32.0 | 33.2 | **77.4** | 28.3 | 42.9 | 49.5 |
| Token-level Entropy | 13.8/10.9 | 48.6 | 74.2 | 33.1 | 34.1 | 35.8 | 77.0 | 30.8 | 37.2 | 48.3 |
| Sentence-level Entropy | 9.6/9.9 | 45.6 | 73.8 | 32.4 | 34.5 | 34.3 | 76.9 | 28.3 | 39.8 | 48.3 |
| **TRAPO (ours)** | 13.8/**13.6** | **51.4** | **79.8** | 33.8 | **40.0** | **38.7** | 77.2 | **35.4** | **43.6** | **52.1** |
| Fully Supervised w/ *8K* Labels | 16.0/12.1 | 52.9 | 78.8 | 36.8 | 42.8 | 39.9 | 77.0 | 29.3 | 42.7 | 49.7 |

Table 9: Overall performance on nine competition-level benchmarks for Qwen-2.5-7B under different top-p settings, with fixed $\Gamma$ (0.5) and a fixed warmup length (5). Training was performed with 1K labeled and 3K unlabeled samples.

| Model | AIME 24/25 | AMC | MATH-500 | Minerva | Olympiad | *Avg.* | ARC-c | GPQA* | MMLU-Pro | *Avg.* |
|---|---|---|---|---|---|---|---|---|---|---|
| Qwen-Base | 11.5/4.9 | 31.3 | 43.6 | 7.4 | 15.6 | 19.0 | 18.2 | 11.1 | 16.9 | 15.4 |
| Qwen-Instruct | 12.5/10.2 | 48.5 | 80.4 | 32.7 | 41.0 | 37.6 | 70.3 | 24.7 | 34.1 | 43.0 |
| top-p = 0.1 | | | | | | | | | | |
| TRAPO | 17.9/13.8 | 58.7 | 81.4 | 38.2 | 45.5 | 42.6 | 83.7 | 37.9 | 46.8 | 56.1 |
| top-p = 0.3 | | | | | | | | | | |
| TRAPO | 16.6/15.7 | 56.0 | 82.6 | 35.6 | 44.0 | 41.7 | 79.7 | 34.3 | 46.7 | 53.6 |
| top-p = 0.5 | | | | | | | | | | |
| TRAPO | 15.9/9.5 | 52.7 | 79.0 | 34.2 | 39.9 | 38.5 | 73.2 | 32.7 | 45.6 | 50.5 |
| top-p = 0.7 | | | | | | | | | | |
| TRAPO | 14.9/10.8 | 53.4 | 81.8 | 34.9 | 41.8 | 39.6 | 75.4 | 36.9 | 43.8 | 52.0 |
| top-p = 1.0 | | | | | | | | | | |
| TRAPO | 14.9/10.7 | 55.3 | 77.8 | 33.1 | 43.6 | 39.2 | 72.6 | 35.4 | 42.7 | 50.2 |

Table 10: Overall performance across nine competition-level benchmarks for Qwen-2.5-7B under varying $\Gamma$ values, with fixed top-p (0.1) and warmup length (5). Training was conducted with 1K labeled samples and 3K unlabeled samples.

| Model | AIME 24/25 | AMC | MATH-500 | Minerva | Olympiad | *Avg.* | ARC-c | GPQA* | MMLU-Pro | *Avg.* |
|---|---|---|---|---|---|---|---|---|---|---|
| Qwen-Base | 11.5/4.9 | 31.3 | 43.6 | 7.4 | 15.6 | 19.0 | 18.2 | 11.1 | 16.9 | 15.4 |
| Qwen-Instruct | 12.5/10.2 | 48.5 | 80.4 | 32.7 | 41.0 | 37.6 | 70.3 | 24.7 | 34.1 | 43.0 |
| $\Gamma = 0.1$ | | | | | | | | | | |
| TRAPO | 15.7/10.9 | 52.6 | 81.1 | 34.5 | 41.3 | 39.4 | 74.0 | 37.1 | 43.2 | 51.4 |
| $\Gamma = 0.3$ | | | | | | | | | | |
| TRAPO | 16.5/12.9 | 56.8 | 81.9 | 37.6 | 45.9 | 41.9 | 81.9 | 38.1 | 46.3 | 55.4 |
| $\Gamma = 0.5$ | | | | | | | | | | |
| TRAPO | 17.9/13.8 | 58.7 | 81.4 | 38.2 | 45.5 | 42.6 | 83.7 | 37.9 | 46.8 | 56.1 |
| $\Gamma = 0.7$ | | | | | | | | | | |
| TRAPO | 14.3/12.7 | 53.9 | 79.2 | 35.1 | 42.6 | 39.6 | 80.6 | 35.6 | 43.7 | 53.3 |
| $\Gamma = 1.0$ | | | | | | | | | | |
| TRAPO | 14.9/13.3 | 53.9 | 79.7 | 34.7 | 42.1 | 39.8 | 81.3 | 35.9 | 43.4 | 53.5 |

Table 11: Overall performance across nine competition-level benchmarks for Qwen-2.5-7B under varying warmup lengths, with fixed top-p (0.1) and fixed $\Gamma$ (0.5). Training conducted with 1K labeled and 3K unlabeled samples.

| Model | AIME 24/25 | AMC | MATH-500 | Minerva | Olympiad | *Avg.* | ARC-c | GPQA* | MMLU-Pro | *Avg.* |
|---|---|---|---|---|---|---|---|---|---|---|
| Qwen-Base | 11.5/4.9 | 31.3 | 43.6 | 7.4 | 15.6 | 19.0 | 18.2 | 11.1 | 16.9 | 15.4 |
| Qwen-Instruct | 12.5/10.2 | 48.5 | 80.4 | 32.7 | 41.0 | 37.6 | 70.3 | 24.7 | 34.1 | 43.0 |
| warm-up length = 2 | | | | | | | | | | |
| TRAPO | 16.1/12.0 | 54.9 | 77.8 | 34.0 | 40.2 | 39.2 | 78.5 | 33.2 | 41.5 | 51.1 |
| warm-up length = 3 | | | | | | | | | | |
| TRAPO | 17.4/13.6 | 57.5 | 80.2 | 37.1 | 43.8 | 41.6 | 81.9 | 36.2 | 44.8 | 54.3 |
| warm-up length = 5 | | | | | | | | | | |
| TRAPO | 17.9/13.8 | 58.7 | 81.4 | 38.2 | 45.5 | 42.6 | 83.7 | 37.9 | 46.8 | 56.1 |
| warm-up length = 8 | | | | | | | | | | |
| TRAPO | 18.2/14.1 | 59.3 | 82.0 | 38.8 | 46.1 | 43.1 | 84.2 | 38.4 | 47.3 | 56.6 |
| warm-up length = 12 | | | | | | | | | | |
| TRAPO | 17.6/13.5 | 58.1 | 80.9 | 37.7 | 44.9 | 42.1 | 83.1 | 37.6 | 46.1 | 55.6 |

Table 12: Overall performance of Qwen2.5-Math-7B under different training sample sizes and annotation ratios.

| Model | In-Distribution Performance | | | | | | Out-of-Distribution Performance | | | |
|---|---|---|---|---|---|---|---|---|---|---|
| | AIME 24/25 | AMC | MATH-500 | Minerva | Olympiad | Avg. | ARC-c | GPQA* | MMLU-Pro | Avg. |
| Original Models | | | | | | | | | | |
| Qwen-Base | 11.5/4.9 | 31.3 | 43.6 | 7.4 | 15.6 | 19.0 | 18.2 | 11.1 | 16.9 | 15.4 |
| Qwen-Instruct | 12.5/10.2 | 48.5 | 80.4 | 32.7 | 41.0 | 37.6 | 70.3 | 24.7 | 34.1 | 43.0 |
| TRAPO Trained on Varying Sample Sizes (12.5% Labeled) | | | | | | | | | | |
| TRAPO w/ 1K Samples | 13.5/10.1 | 52.3 | 80.7 | 39.4 | 42.2 | 39.7 | 75.2 | 24.1 | 43.5 | 47.6 |
| TRAPO w/ 2K Samples | 15.0/11.6 | 53.3 | 81.2 | 38.9 | 44.2 | 40.7 | 82.4 | 28.7 | 45.2 | 52.1 |
| TRAPO w/ 4K Samples | 16.1/12.9 | 56.8 | 82.3 | 36.7 | 45.4 | 41.7 | 82.1 | 33.8 | 46.7 | 54.2 |
| TRAPO w/ 16K Samples | 21.3/16.1 | 60.9 | 84.8 | 38.2 | 43.3 | 44.1 | 82.6 | 39.5 | 46.2 | 56.1 |
| TRAPO Trained on Varying Sample Sizes (25% Labeled) | | | | | | | | | | |
| TRAPO w/ 1K Samples | 17.1/12.8 | 53.6 | 79.4 | 39.3 | 41.5 | 40.6 | 72.7 | 30.3 | 42.4 | 48.5 |
| TRAPO w/ 2K Samples | 18.1/14.3 | 55.4 | 81.6 | 33.1 | 43.4 | 41.0 | 82.6 | 39.4 | 45.0 | 55.7 |
| TRAPO w/ 4K Samples | 17.9/13.8 | 58.7 | 81.4 | 38.2 | 45.5 | 42.6 | 83.7 | 37.9 | 46.8 | 56.1 |
| TRAPO w/ 16K Samples | 24.3/17.1 | 60.0 | 84.6 | 39.3 | 48.3 | 45.6 | 84.6 | 43.9 | 50.7 | 59.7 |
| TRAPO Trained on Varying Sample Sizes (50% Labeled) | | | | | | | | | | |
| TRAPO w/ 1K Samples | 14.3/10.9 | 51.7 | 81.4 | 34.2 | 42.1 | 39.1 | 78.3 | 30.1 | 45.2 | 51.2 |
| TRAPO w/ 2K Samples | 16.2/13.1 | 54.8 | 82.3 | 37.1 | 45.7 | 41.5 | 81.5 | 34.2 | 46.6 | 54.1 |
| TRAPO w/ 4K Samples | 17.3/15.7 | 59.2 | 83.9 | 39.4 | 47.3 | 43.8 | 83.7 | 36.8 | 46.6 | 55.7 |
| TRAPO w/ 16K Samples | 24.4/18.3 | 61.5 | 84.1 | 40.8 | 46.3 | 45.9 | 84.2 | 43.7 | 49.7 | 59.2 |
| Fully Supervised w/ 45K Labels | 25.1/15.3 | 62.0 | 84.4 | 39.3 | 46.8 | 45.5 | 82.3 | 40.4 | 49.3 | 57.3 |

Table 13: Qwen-2.5-7B results on nine competition-level benchmarks using 1K labeled and 3K unlabeled samples (30% reliable data selected by Sentence-level Entropy, Self-certainty, and TRAPO).

| Model | AIME 24/25 | AMC | MATH-500 | Minerva | Olympiad | Avg. | ARC-c | GPQA* | MMLU-Pro | Avg. |
|---|---|---|---|---|---|---|---|---|---|---|
| Qwen-Base | 11.5/4.9 | 31.3 | 43.6 | 7.4 | 15.6 | 19.0 | 18.2 | 11.1 | 16.9 | 15.4 |
| Qwen-Instruct | 12.5/10.2 | 48.5 | 80.4 | 32.7 | 41.0 | 37.6 | 70.3 | 24.7 | 34.1 | 43.0 |
| Random | 15.8/12.3 | 53.5 | 79.8 | 34.8 | 41.8 | 39.7 | 80.8 | 35.8 | 43.2 | 53.3 |
| Sentence-level Entropy | 16.3/12.5 | 54.6 | 80.2 | 35.3 | 42.4 | 40.2 | 81.8 | 35.4 | 43.7 | 53.6 |
| Self-certainty | 15.8/13.3 | 52.9 | 80.7 | 36.6 | 43.5 | 40.5 | 80.4 | 35.8 | 42.9 | 53.0 |
| TRAPO | 16.7/13.7 | 57.1 | 81.0 | 37.3 | 44.6 | **41.8** | 83.2 | 37.4 | 45.9 | **55.5** |

Table 14: Overall performance on nine competition-level benchmarks for Qwen-2.5-7B using random selection or TRAPO. Training was conducted with 1K labeled samples and 3K unlabeled samples.

| Model | AIME 24/25 | AMC | MATH-500 | Minerva | Olympiad | Avg. | ARC-c | GPQA* | MMLU-Pro | Avg. |
|---|---|---|---|---|---|---|---|---|---|---|
| Qwen-Base | 11.5/4.9 | 31.3 | 43.6 | 7.4 | 15.6 | 19.0 | 18.2 | 11.1 | 16.9 | 15.4 |
| Qwen-Instruct | 12.5/10.2 | 48.5 | 80.4 | 32.7 | 41.0 | 37.6 | 70.3 | 24.7 | 34.1 | 43.0 |
| No Selection | 14.2/13.5 | 52.6 | 80.2 | 34.9 | 40.9 | 39.4 | 76.2 | 36.4 | 43.6 | 52.1 |
| 10% Selected | | | | | | | | | | |
| Random | 14.9/13.3 | 53.9 | 79.7 | 34.7 | 42.1 | 39.8 | 80.3 | 34.9 | 42.4 | 52.5 |
| TRAPO | 15.8/13.5 | 55.0 | 80.3 | 35.8 | 43.2 | 40.7 | 81.5 | 35.8 | 43.5 | 53.6 |
| 30% Selected | | | | | | | | | | |
| Random | 15.8/12.3 | 53.5 | 79.8 | 34.8 | 41.8 | 39.7 | 80.8 | 35.8 | 43.2 | 53.3 |
| TRAPO | 16.7/13.7 | 57.1 | 81.0 | 37.3 | 44.6 | 41.8 | 83.2 | 37.4 | 45.9 | 55.5 |
| 50% Selected | | | | | | | | | | |
| Random | 14.5/12.8 | 51.5 | 77.2 | 31.5 | 40.0 | 37.9 | 77.8 | 34.8 | 41.8 | 51.5 |
| TRAPO | 15.1/13.6 | 54.2 | 80.5 | 35.2 | 42.5 | 40.1 | 82.0 | 36.2 | 43.8 | 54.0 |
| 70% Selected | | | | | | | | | | |
| Random | 14.6/13.0 | 52.4 | 78.5 | 34.0 | 40.8 | 38.4 | 79.2 | 35.2 | 42.5 | 52.3 |
| TRAPO | 14.9/13.5 | 53.8 | 79.9 | 34.9 | 41.9 | 39.8 | 81.0 | 35.4 | 42.8 | 53.1 |
| All Selection | 14.9/10.7 | 55.3 | 77.8 | 33.1 | 43.6 | 39.2 | 72.6 | 35.4 | 42.7 | 50.2 |

Table 15: Overall performance across nine competition-level benchmarks for Qwen-2.5-7B with varying ratios ($\sigma_M$) of unlabeled samples. Training uses 1K labeled samples and 3K unlabeled samples.

| Model | AIME 24/25 | AMC | MATH-500 | Minerva | Olympiad | *Avg.* | ARC-c | GPQA* | MMLU-Pro | *Avg.* |
|---|---|---|---|---|---|---|---|---|---|---|
| Qwen-Base | 11.5/4.9 | 31.3 | 43.6 | 7.4 | 15.6 | 19.0 | 18.2 | 11.1 | 16.9 | 15.4 |
| Qwen-Instruct | 12.5/10.2 | 48.5 | 80.4 | 32.7 | 41.0 | 37.6 | 70.3 | 24.7 | 34.1 | 43.0 |
| *$\sigma_M = 0.00$* | 14.2/13.5 | 52.6 | 80.2 | 34.9 | 40.9 | 39.4 | 76.2 | 36.4 | 43.6 | 52.1 |
| *$\sigma_M = 0.25$* | | | | | | | | | | |
| Token-level Entropy | 16.7/13.6 | 54.6 | 81.4 | 34.3 | 41.3 | 40.4 | 79.6 | 35.9 | 44.6 | 53.4 |
| TRAPO | 14.6/13.6 | 55.4 | 79.8 | 35.7 | 42.1 | 40.2 | 81.9 | 35.4 | 44.0 | 53.8 |
| *$\sigma_M = 0.50$* | | | | | | | | | | |
| Token-level Entropy | 15.0/12.4 | 51.6 | 79.8 | 32.7 | 39.9 | 38.6 | 77.3 | 34.8 | 42.9 | 51.7 |
| TRAPO | 16.8/13.6 | 56.2 | 80.5 | 38.9 | 43.6 | 41.6 | 82.8 | 36.6 | 44.9 | 54.8 |
| *$\sigma_M = 0.75$* | | | | | | | | | | |
| Token-level Entropy | 16.2/13.4 | 52.1 | 79.0 | 33.8 | 39.1 | 38.9 | 77.6 | 29.8 | 41.3 | 49.6 |
| TRAPO | 17.4/12.9 | 57.2 | 80.8 | 37.5 | 44.3 | 41.7 | 82.5 | 37.2 | 45.9 | 55.2 |
| *$\sigma_M = 1.00$* | | | | | | | | | | |
| Token-level Entropy | 18.2/11.9 | 53.4 | 80.2 | 34.6 | 41.9 | 40.0 | 72.9 | 32.3 | 44.0 | 49.7 |
| TRAPO | 17.9/13.8 | 58.7 | 81.4 | 38.2 | 45.5 | 42.6 | 83.7 | 37.9 | 46.8 | 56.1 |

Table 16: Overall performance across nine competition-level benchmarks for Qwen-2.5-7B, averaged over three runs. Training was performed with 1K labeled samples and 3K unlabeled samples.

| Model | AIME 24/25 | AMC | MATH-500 | Minerva | Olympiad | *Avg.* | ARC-c | GPQA* | MMLU-Pro | *Avg.* |
|---|---|---|---|---|---|---|---|---|---|---|
| Qwen-Base | 11.5/4.9 | 31.3 | 43.6 | 7.4 | 15.6 | 19.0 | 18.2 | 11.1 | 16.9 | 15.4 |
| Qwen-Instruct | 12.5/10.2 | 48.5 | 80.4 | 32.7 | 41.0 | 37.6 | 70.3 | 24.7 | 34.1 | 43.0 |
| TRAPO | $18.2 \pm 0.3$ / $13.6 \pm 0.2$ | $59.3 \pm 0.5$ | $81.9 \pm 0.4$ | $37.9 \pm 0.4$ | $45.8 \pm 0.5$ | $42.8 \pm 0.4$ | $83.9 \pm 0.6$ | $37.8 \pm 0.6$ | $47.5 \pm 0.5$ | $56.4 \pm 0.5$ |

Table 17: Qwen-2.5-7B results on nine competition-level benchmarks using 1K labeled and 3K unlabeled samples, with average trajectory matching or maximum trajectory matching.

| Model | AIME 24/25 | AMC | MATH-500 | Minerva | Olympiad | *Avg.* | ARC-c | GPQA* | MMLU-Pro | *Avg.* |
|---|---|---|---|---|---|---|---|---|---|---|
| Qwen-Base | 11.5/4.9 | 31.3 | 43.6 | 7.4 | 15.6 | 19.0 | 18.2 | 11.1 | 16.9 | 15.4 |
| Qwen-Instruct | 12.5/10.2 | 48.5 | 80.4 | 32.7 | 41.0 | 37.6 | 70.3 | 24.7 | 34.1 | 43.0 |
| TRAPO-MAX | 16.3/9.9 | 52.7 | 80.8 | 35.6 | 41.3 | 39.4 | 81.6 | 33.2 | 42.6 | 52.5 |
| TRAPO-MEAN | 17.9/13.8 | 58.7 | 81.4 | 38.2 | 45.5 | 42.6 | 83.7 | 37.9 | 46.8 | 56.1 |

