# OpenReview forum: "TraPO: A Semi-Supervised Reinforcement Learning Framework for Boosting LLM Reasoning"
_ICLR.cc/2026/Conference — ICLR 2026 Poster_

### Official Review · Reviewer_NfpB · 2025-10-24

**Soundness:** 3
**Presentation:** 3
**Contribution:** 2
**Rating:** 4
**Confidence:** 3

**Summary:**

TRAPO tackles semi-supervised RL with verifiable rewards (RLVR) for LLM reasoning. Core idea: use a small labeled set to **anchor** training while selecting reliable unlabeled items via **trajectory similarity**—match each unlabeled sample’s pass-rate curve to the averaged labeled trajectory, then include only top-p/above-threshold items in GRPO updates. Experiments on math ID (AIME/AMC/Minerva/MATH-500/Olympiad) and OOD (ARC-c/GPQA*/MMLU-Pro) show notable label-efficiency gains (e.g., 1k L + 3k U outperforming stronger unsupervised methods; 4k L + 12k U rivaling/ surpassing fully supervised on 45k L). Includes an informal generalization bound tying trajectory consistency to target risk.

**Strengths:**

1. **Clean, intuitive mechanism:** “learn from how learning evolves,” not from point estimates. The trajectory-matching mask is simple to implement atop GRPO and robust to domain shift.
2. **Label efficiency & breadth:** Strong results with tiny labeled sets; both ID and OOD improvements; ablations with OOD unlabeled data are convincing.
. **Theoretical framing:** A readable (if high-level) bound linking trajectory similarity and confidence to generalization; clarifies the role of labeled anchors.

**Weaknesses:**

1. **Novelty positioning:** Semi-supervised filtering by “learning dynamics” echoes curriculum/confidence-based selection; paper could contrast more sharply vs. entropy/self-certainty and preference-based filtering beyond empirical gains.
2. **Proxy fidelity:** Unlabeled “pseudo-pass rates” depend on majority voting; risk of reinforcing easy/short answers remains. Need stress tests where voting is systematically biased.
3. **Sensitivity & cost:** Top-p/Γ thresholds, warm-up length, rollouts (G=8), batch sizes, β=0, entropy coef=0.01 with no thorough sweep or compute-normalized comparison; H200×8 requirements make label-efficiency claims less practical if compute rises.
4. **Theory scope:** Bound is informative but rests on simplifying assumptions (e.g., confidence terms, domain discrepancy decomposition); no empirical diagnostics linking bound terms to measured quantities.
5. **Generalization breadth:** Mostly math-centric; limited non-math/OOD beyond three benchmarks; no noisy-label or adversarial-feedback scenarios.

**Questions:**

Please address the weakness above.

---

> ### Author Response · Authors · 2025-11-21
>
> **Response to Reviewer NfpB:**
> Thanks for your comments! Below we address the feedback and comments in detail:
>
> >**W1:** Novelty positioning: Semi-supervised filtering by ``learning dynamics'' echoes curriculum/confidence-based selection; paper could contrast more sharply vs. entropy/self-certainty and preference-based filtering beyond empirical gains.
> >**A:**
> Under a fixed selection ratio of 30%, we compare TraPO to **random selection**, **sentence-level entropy** selection (lower entropy indicates more reliable pseudo-labels), and **self-certainty** (higher self-certainty indicates reliability). The experimental results in the table below show that TraPO consistently outperforms all others on both ID and OOD test sets, highlighting its robust effectiveness. For comprehensive results, see Table 13 in the appendix.
> Notably, TraPO selects unlabeled examples whose learning dynamics during training resemble those of labeled data. Unlike traditional entropy- or confidence-based filtering, TraPO emphasizes **the trend of model improvement over time, which naturally excludes noisy or trivial samples**: unlabeled samples that are too hard tend to remain near zero, while those that are too easy show an early spike but do not improve significantly as training progresses.
>
>
> | Model                  | ID Avg. | OOD Avg. |
> |------------------------|---------|-----------|
> | Qwen-Base              | 19.0    | 15.4      |
> | Qwen-Instruct          | 37.6    | 43.0      |
> | Random                 | 39.7    | 53.3      |
> | Sentence-level Entropy | 40.2    | 53.6      |
> | Self-certainty         | 40.5    | 53.0      |
> | TraPO                  | 41.8    | 55.5      |
>
>
> >**W2:** Proxy fidelity: Unlabeled "pseudo-pass rates" depend on majority voting; risk of reinforcing easy/short answers remains. Need stress tests where voting is systematically biased.
> **A:** Thanks for the comment.
> First, our method **does not excessively reinforce easy problems** because our goal with trajectory matching is to identify unlabeled examples that behave like labeled ones during training. Their pass rates rise steadily as the model improves. In contrast, examples that are too easy show an early spike but quickly stop improving, which does not meet our selection criteria.
> Second, our method **does not excessively reinforce short answers**. Our selection does not favor simple problems, and after training with TraPO, the average output length of the model increased from around **1,000** to about **2,000**. This also serves as supporting evidence.
> Finally, we're sorry if we've overlooked something, but as far as we know, there haven't really been stress testing experiments like this before. So we're honestly not quite sure how to design such experiments. Could you provide us with more specific requirements or suggestions? Since there is still time for the rebuttal, we will do our best to accommodate your experimental requests.

---

> ### Author Response · Authors · 2025-11-21
>
> >**W3-1:** Sensitivity & cost: Top-p/$\Gamma$ thresholds, warm-up length, rollouts (G=8), batch sizes, $\beta$=0, entropy coef=0.01 with no thorough sweep or compute-normalized comparison.
> >**A:** Thank you for your suggestions. We conduct sensitivity analyses on our main hyperparameters: top-p, $\Gamma$, and warm-up length. Each experiment costs about 537.6 dollars and takes about a day, so we are unable to analyze other regular RLVR hyperparameters, such as the number of rollouts (G = 8), batch size, $\beta$ = 0, or the entropy coefficient = 0.01. For these, we use the original baseline [1] settings to enable fair comparison. We present the overall experimental results below. **For detailed results, please refer to Tables 9, 10, and 11 in the appendix of the updated paper**.
> For **top-p**, we find that larger values introduce noisier responses early in training, which degrade model performance due to unreliable pseudo-labels.
> For $\Gamma$, setting it too low (e.g., 0.1) admits too many low-quality unlabeled samples, while setting it too high (e.g., above 0.7) is overly conservative and underutilizes the unlabeled data. Both extremes reduce performance.
> As for **warm-up length**, a very short warm-up (e.g., 2 epochs) yields unreliable estimates when unlabeled samples are first introduced, hurting performance. Performance improves and stabilizes as warm-up length increases. We chose 5 epochs as a balance, long enough for stable pseudo-label estimation, yet early enough to start leveraging unlabeled data efficiently.
>
> | Top-p | 0.1 | 0.3 | 0.5 | 0.7 | 1.0 |
> |----------------|-----|-----|-----|-----|------|
> | TraPO (ID Avg.)  | 42.6 | 41.7 | 38.5 | 39.6 | 39.2 |
> | TraPO (OOD Avg.) | 56.1 | 53.6 | 50.5 | 52.0 | 50.2 |
>
> | $\Gamma$ | 0.1 | 0.3 | 0.5 | 0.7 | 1.0 |
> |----------------|-----|-----|-----|-----|------|
> | TraPO (ID Avg.)  | 39.4 | 41.9 | 42.6 | 39.6 | 39.8 |
> | TraPO (OOD Avg.) | 51.4 | 55.4 | 56.1 | 53.3 | 53.5 |
>
> | Warm-up Length | 2   | 3   | 5   | 8   | 12  |
> |-------------------------|-----|-----|-----|-----|-----|
> | TraPO (ID Avg.)  | 39.2 | 41.6 | 42.6 | 43.1 | 42.1 |
> | TraPO (OOD Avg.) | 51.1 | 54.3 | 56.1 | 56.6 | 55.6 |
>
>
> >**W3-2:** H200×8 requirements make label-efficiency claims less practical if compute rises.
> >**A:** First, we'd like to clarify that the type of GPU used relates to computational resource requirements. This is not related to our contribution on label efficiency, where we achieve fully supervised performance using only **10%** of the labeled data.
> Second, because large-scale RLVR tasks require substantial GPU resources, it is **common** to use high-performance GPUs such as the A100, H20, or H200 [1-4]. However, our method is not limited to these particular hardware models. It can be implemented on different GPUs as long as the hyperparameters, such as batch size, are appropriately adjusted to match different memory capacities.
>
>
> >**W4:** Theory scope: Bound is informative but rests on simplifying assumptions (e.g., confidence terms, domain discrepancy decomposition); no empirical diagnostics linking bound terms to measured quantities.
> >**A:**  In Figure 4, we already observe an empirical trend consistent with our theoretical bound: as the Trajectory Cosine Similarity (TCS) increases, the error in pseudo-label estimation decreases, supporting the role of TCS as a reliability indicator in the bound.
> However, due to computational constraints, the more detailed quantitative analysis linking each term of the bound to measurable quantities is beyond the scope of our current study. We plan to investigate these relationships thoroughly as part of future work.
> That said, **we view the theoretical framework primarily as a guiding principle for algorithm design**. It motivates our use of trajectory alignment and trend-based filtering in TraPO, which in turn leads to the empirical gains reported in the paper.

---

> ### Author Response · Authors · 2025-11-21
>
> >**W5-1:** Generalization breadth: Mostly math-centric; limited non-math/OOD beyond three benchmarks.
> >**A:**
> In fact, the MMLU-PRO dataset used in our evaluation includes test samples from **14 diverse domains**. These domains include **Business, Biology, Philosophy, Health, History, Law, Economics, Physics, Chemistry, etc**. Therefore, our experiments demonstrate TraPO's generalization ability across a broad range of out-of-distribution subjects, not just three. This breadth motivates future work to further investigate and enhance TraPO's robustness across even more diverse and challenging domains.
>
> >**W5-2:** No noisy-label or adversarial-feedback scenarios.
> >**A:**   As the first semi-supervised RLVR training framework, **our main contribution is to demonstrate that semi-supervised RLVR is practical and effective**. We achieve high label efficiency, with only **4K** labeled samples outperforming previous results with **45K** labeled examples, and validate our approach through multiple experiments under standard semi-supervised learning assumptions.
> Although scenarios with noisy labels or adversarial feedback are important, **they are beyond the scope of this initial exploratory work**. We will investigate them in greater depth in future research.
>
>
> >**Claims on Some Weaknesses:**
> Finally, we'd like to sincerely engage in discussion with the reviewer. We have observed that some of the weaknesses raised may not fully reflect the realities of LLM research. Unlike traditional classification tasks such as CIFAR or ImageNet, which have low computational demands and closely match mathematical assumptions, LLM reasoning does not fit a simple classification framework and involves extremely large models with consequently high computational costs. Even now, LLM researchers cannot supplement experiments as comprehensively as is possible with CIFAR, and theoretical analysis of LLMs is still in its early stages. More importantly, there has been little exploration in the community regarding the potential for semi-supervised RLVR on LLMs and our work is the first of its kind.
> We don't attempt to claim our paper is perfect. We'd like to highlight that, given our limited resources, we have done our utmost to provide insights in algorithm design, experimentation, and theoretical analysis for this initial exploration of semi-supervised RLVR. It is true that for the reasons mentioned, we could not cover every aspect such as the full sensitivity analysis of all parameters, provide a complete mathematical explanation for LLMs, or examine topics like noisy-label or adversarial scenarios that have not reached consensus in the community. Still, we firmly believe these are not fundamental weaknesses of our work. We sincerely hope the reviewer will recognize the value of our contributions. Thank you!
>
>
> [1] Yan J, Li Y, Hu Z, et al. Learning to reason under off-policy guidance[C]. NeurIPS'25
> [2] Zhao X, Kang Z, Feng A, et al. Learning to reason without external rewards[J]. arXiv preprint arXiv:2505.19590, 2025.
> [3] Zuo Y, Zhang K, Sheng L, et al. Ttrl: Test-time reinforcement learning[J]. arXiv preprint arXiv:2504.16084, 2025.
> [4] Agarwal S, Zhang Z, Yuan L, et al. The unreasonable effectiveness of entropy minimization in llm reasoning[J]. arXiv preprint arXiv:2505.15134, 2025.

---

> ### Author Response · Authors · 2025-11-25
>
> Dear Reviewer NfpB,
>
> We hope this message finds you well.
>
> Once again, we sincerely appreciate your valuable time and thoughtful feedback. We are reaching out to kindly remind you that, based on your suggestions, we have included additional experiments and discussions in our response.
> If you could take valuable time out of your schedule to review our reply, we would be greatly appreciative. Your timely feedback will help us confirm whether we have appropriately addressed the issues and suggestions you raised.
>
> Furthermore, your insights and professional expertise are crucial to us. If there are any further areas where we can enhance the quality of the paper, please let us know. We are more than willing to address any remaining concerns.
>
> Warmest regards,
> The Authors

---

> > ### Comment · Reviewer_NfpB · 2025-11-25
> >
> > Thank you for the response, some of concerns are addressed. I will wait until the end of the discussion period to make the final decision about my score.

---

> > > ### Author Response · Authors · 2025-11-26
> > >
> > > Thanks for your response. While the discussion is ongoing, we would be keen to learn what, in your opinion, still needs to be amended in our paper. Please do not hesitate to let us know. We look forward to your further feedback and are very willing to address any remaining concerns.
> > >
> > > Best regards,
> > > Authors

---

### Official Review · Reviewer_aiJH · 2025-10-24

**Soundness:** 2
**Presentation:** 3
**Contribution:** 2
**Rating:** 4
**Confidence:** 4

**Summary:**

This paper presents TRAPO, a semi-supervised RLVR paradigm. TRAPO measures the similarity between trajectories from unlabeled and labeled data across training epochs and updates the model using only reliable samples. Experiments on the Qwen2.5-7B model demonstrate that TRAPO achieves higher accuracy across several math reasoning benchmarks under limited labeled data settings, and in some cases even surpasses models trained with fully labeled data.

**Strengths:**

- Tackles an important and practical problem in semi-supervised RLVR.
- The proposed approach is overall reasonable and well-motivated.
- The paper is clearly written and well-organized.
- Experimental results show promising improvements.

**Weaknesses:**

- Some design choices require stronger justification (see below).
- Missing key ablation studies to verify and better understand design components.

**Questions:**

Thank you for submitting this interesting and well-written paper. The problem is timely and relevant, and the proposed direction is promising. Below are my specific questions and suggestions for improvement:

- Q1: Would not the ordering of samples within each epoch affect the results? Because the trajectories of each sample highly depends on the model's current capability.

- Q2: Why is the average trajectory used as a proxy for similarity a good choice? Have the authors considered clustering the unlabeled data first and then computing per-cluster averages for more robust similarity estimation?

- Q3: Since TRAPO evaluates all data each epoch but only updates with part of it, what is the training overhead, and how does it scale with dataset size?

- Q4: Could the authors clarify the expected-answer criteria used in evaluation (e.g., is $a_i = y$ for all $q \in D_u$)?

- Q5: The evaluation should include more models. While the Llama-3.1 experiments are appreciated, they lack fair baselines comparable to those in Qwen.

- Q6: It would be informative to report results under different labeled data sizes (e.g., 10K, 20K).

- Q7: Please include a sensitivity analysis for hyperparameters $p$ and threshold $T$ to better understand their influence on performance.

---

> ### Author Response · Authors · 2025-11-21
>
> **Response to Reviewer aiJH:**
> Thank you for your thoughtful comments and helpful suggestions! They have been incredibly valuable to our work. We apologize for the delay in our response; limited computational resources for LLM training mean that each experiment takes quite a while to complete. We are pleased, however, that we have been able to address most of the reviewers' questions positively.
> Below are our detailed responses to your comments:
>
>
> > **Q1:**
> Would not the ordering of samples within each epoch affect the results? Because the trajectories of each sample highly depends on the model's current capability.
> >**A:** Thank you for this thoughtful question.
> First, we want to clarify that **our implementation does indeed fully shuffle the data within each epoch**, just like all baseline methods. This demonstrates that our results are not sensitive to the sample order.
> Second, TraPO selects the core samples **at each minibatch step** by evaluating a fixed number (e.g., 128) of unlabeled samples within the batch. It compares their trajectories with those in the reliable database and selects a subset (e.g., 10%-30% of 128) for training. **We do not evaluate the entire unlabeled pool, as this would obviously be infeasible**. In this process, all minibatch samples are evaluated using the same model checkpoint to ensure a fair comparison. For clarity, we have included a schematic diagram of the TraPO framework in the updated paper; **please refer to Figure 3**.
> Finally, TraPO empirically demonstrates minimal dependence on sample order. In three trials using shuffled data (Qwen-2.5-7B, 1K labeled, 3K unlabeled), both the results and sample selections were nearly identical, confirming the method's robustness (see table below).
>
>
> **Overall performance across nine competition-level benchmarks for Qwen-2.5-7B (averaged over three runs). Training used 1K labeled + 3K unlabeled samples.**
>
> | Model | AIME 24/25 | AMC | MATH-500 | Minerva | Olympiad | Avg. | ARC-c | GPQA* | MMLU-Pro | Avg. |
> |-------|------------|-----|----------|---------|-----------|------|--------|--------|-----------|--------|
> | **Qwen-Base** | 11.5 / 4.9 | 31.3 | 43.6 | 7.4 | 15.6 | 19.0 | 18.2 | 11.1 | 16.9 | 15.4 |
> | **Qwen-Instruct** | 12.5 / 10.2 | 48.5 | 80.4 | 32.7 | 41.0 | 37.6 | 70.3 | 24.7 | 34.1 | 43.0 |
> | **TraPO** | 18.2 ± 0.3 / 13.6 ± 0.2 | 59.3 ± 0.5 | 81.9 ± 0.4 | 37.9 ± 0.4 | 45.8 ± 0.5 | 42.8 ± 0.4 | 83.9 ± 0.6 | 37.8 ± 0.6 | 47.5 ± 0.5 | 56.4 ± 0.5 |

---

> ### Author Response · Authors · 2025-11-21
>
> > **Q2:**
> Why is the average trajectory used as a proxy for similarity a good choice? Have the authors considered clustering the unlabeled data first and then computing per-cluster averages for more robust similarity estimation?
> >**A:**
> **(1) Clustering is infeasible for LLM training**
> In fact, clustering would require extracting response representations for all unlabeled samples and running iterative clustering algorithms, making the process both computationally expensive and less scalable.
> Furthermore, large language models in RLVR tasks differ fundamentally from traditional semi-supervised settings. First, the **semantic independence** between samples makes it impractical to establish relationships among their answers through clustering. Second, extracting sample representations from large language models is **time-consuming**, rendering online clustering during RLVR training prohibitively expensive. As a result, clustering-based approaches have not been applied to RLVR tasks.
> This distinction explains why traditional semi-supervised methods fail in RLVR, whereas our approach, TraPO, successfully bridges labeled and unlabeled samples using pass rates. Thus, **TraPO establishes the first semi-supervised RLVR framework.**
> **(2) Why is using the average trajectory a good choice?**
> As we noted in our manuscript, our goal with average trajectory matching is to **identify unlabeled examples that exhibit training behaviors similar to those of labeled samples**. Let's examine how different data samples behave during training:
> First, the average pass rates of labeled samples usually show **steadily increasing trends as the model learns**. Second, those unlabeled samples that are unlearnable, possibly because the LLM neither acquired their patterns during pretraining nor can infer them from the labeled data, **tend to remain near zero in pass rate**. Conversely, those unlabeled samples that are too easy **exhibit early spikes** in pass rate but show little improvement over time.
> Importantly, the selected samples tend to display similar learning trajectories and reasoning patterns as the labeled samples. They are **neither too hard**, which would disrupt LLM training because they are unlearnable, **nor too easy**, which could cause overfitting, thus supporting a more effective and balanced training process.
> One may also consider other variants, such as not using the average pass rate trajectory (TraPO-Mean), but instead selecting, from the unlabeled samples, those whose pass rate trajectory is most similar (TraPO-Max) to the trajectory of **any** individual labeled sample for inclusion in training. However, this approach can lead to unstable selection because, among the unlabeled samples, problems that are too difficult, too easy, or of moderate difficulty can **all find relatively similar pass rate trajectories** among the labeled samples. As a result, the selection is not very effective (Table below).  For comprehensive results, please see Table 17 in the appendix.
> In summary, we use the average pass-rate trajectory of labeled data as a proxy for similarity is that it is **simple, fast, and effective in an online setting**.
>
> **Qwen-2.5-7B results on nine competition-level benchmarks using 1K labeled and 3K unlabeled samples,
> with average trajectory matching (TraPO-Mean) or maximum trajectory (TraPO-Max) matching.**
>
> | Model           | ID Avg.  | OOD Avg. |
> |-----------------|-----------|-------|
> | **Qwen-Base**        | 19.0  | 15.4 |
> | **Qwen-Instruct**  | 37.6| 43.0 |
> | **TraPO-Max**       | 39.4 | 52.5 |
> | **TraPO-Mean**     | 42.6  | 56.1 |

---

> ### Author Response · Authors · 2025-11-21
>
> >**Q3:** Since TRAPO evaluates all data each epoch but only updates with part of it, what is the training overhead, and how does it scale with dataset size?
> >**A:**
> Thank you for this important question.
> TraPO's sample selection is applied to each minibatch. At every training step, we evaluate the samples in the minibatch (e.g., 128), compare their pass-rate trajectories against a reference database, and select a subset (e.g., 10%-30% of 128) for training. This approach eliminates the need to process all unlabeled samples at once, which would be impractical for large-scale LLM training. **The only additional operation involved is a cosine similarity computation over short vectors, which is negligible compared to the cost of response generation**.
> TraPO's overhead scales **linearly** with dataset size, similar to standard RLVR [1], and adds negligible cost per epoch. In our experiments, all methods use the same number of epochs, batch size, and hardware (8x H200). The time costs of each paradigm are presented in the table below. These results show that TraPO incurs no substantial additional training cost compared to supervised RLVR.
> Additionally, to explain why the effective duration of unsupervised training is so short: due to the excessive noise during unsupervised training, the model quickly experiences model collapse. We have included the time consumption comparison results in the appendix. Please refer to Table 7 in the updated paper.
>
>
> **Table: Wall-clock training time (reported as "GPU-hours × GPUs") across data regimes**
> | Data Size | Unsupervised (Early Collapse)| Supervised | Semi-Supervised (TraPO) |
> |-----------|--------------|------------|---------------------------|
> | 4K        | ~7 × 8       | ~25 × 8    | ~26 × 8                  |
> | 8K        | ~13 × 8      | ~39 × 8    | ~38 × 8                  |
> | 45K       | ~11 × 8      | ~57 × 8    | ~55 × 8                  |
>
>
> >**Q4:** Could the authors clarify the expected-answer criteria used in evaluation?
> >**A:**
> **For labeled samples**, the expected answer is the dataset's **ground-truth label**.
> **For each unlabeled sample**, we generate G responses and use Math-Verify, a lightweight and widely used conclusion extraction tool, to extract the final answer from each response. The most frequently occurring conclusion is used as the pseudo-label (expected answer) for reward computation, and its frequency divided by G serves as the pass rate. If multiple conclusions tie for the highest count, no reward is assigned for that rollout, and the pass rate is set to 0.
>
> >**Q5:** The evaluation should include more models. While the Llama-3.1 experiments are appreciated, they lack fair baselines comparable to those in Qwen.
> >**A:** Thank you for your valuable suggestions. We supplement the Llama-3.1-8B model with the same comparison baselines as Qwen. The average performance on ID and OOD is shown in the table below; **for detailed experimental results, please refer to Table 5 in the appendix of the paper**. On the Llama-3.1-8B model, we use 1K labeled ID samples and 1K unlabeled OOD samples for experiments. The results show that our TraPO method outperforms all unsupervised baselines and the naive semi-supervised baseline, which highlights the applicability and insensitivity of TraPO across models.
>
>
> | Method | ID Avg. | OOD Avg. |
> |--------|---------|-----------|
> | Original Model Llama-3.1-8B-Instruct | 17.1 | 21.1 |
> | **Unsupervised Methods Trained on 1K Unlabeled ID Samples & 1K Unlabeled OOD Samples** | | |
> | TTRL | 19.5 | 17.6 |
> | Self-certainty | 19.1 | 17.6 |
> | Token-level Entropy | 18.0 | 16.4 |
> | Sentence-level Entropy | 19.3 | 17.7 |
> | **Semi-supervised Methods Trained on 1K Labeled ID Samples & 1K Unlabeled OOD Samples** | | |
> | TTRL | 19.3 | 17.5 |
> | Self-certainty | 19.0 | 17.7 |
> | Token-level Entropy | 18.6 | 17.6 |
> | Sentence-level Entropy | 19.6 | 18.1 |
> | **TraPO (ours)** | 20.7 | 18.5 |
> | **Fully Supervised Training w/ 1K Labeled ID Samples & 1K Labeled OOD Samples** | | |
> | Fully Supervised | 20.2 | 19.3 |

---

> ### Author Response · Authors · 2025-11-21
>
> >**Q6:** It would be informative to report results under different labeled data sizes (e.g., 10K, 20K).
> >**A:**
> Thank you for this insightful suggestion!
> Due to the high computational cost of RLVR training, approximately one day for 4K samples, and limited resources, we regret that we cannot run new experiments with 10K or 20K labeled samples within the rebuttal period.
> That said, your intuition aligns closely with our own follow-up study. After submission, we conducted experiments across varying data scales (**1K, 2K, 4K**, and **16K** total samples) and labeling ratios (**12.5%, 25%**, and **50%**). The overall experimental results are shown in the table below. **For detailed experimental results, please refer to Table 12 in the paper**. Our findings show two key trends:
> First, under a fixed labeling ratio, TraPO's performance **consistently improves as the total dataset size increases**. For clarity, we illustrate this trend (scaling law) in Figure 1 of the updated paper.
> Second, **once the labeling ratio reaches a moderate level, TraPO can match the performance of fully supervised training without requiring additional labels**. For example, with 16K total samples, both 25% and 50% labeling achieve performance close to that of full supervision on 45K labeled samples, with no significant gain from increasing from 25% to 50%. In contrast, a lower ratio, such as 12.5%, yields a lower performance ceiling, likely because the limited labeled set fails to capture the diverse reasoning patterns in the unlabeled data, leading to less stable selection.
> Notably, with only 4K labeled samples (**less than 10% of the full 45K set**), TraPO already outperforms fully supervised training on 45K samples, demonstrating its strong label efficiency.
>
>
> | Data Size | 1K | 2K | 4K | 16K |
> |--------------------|----|----|----|------|
> | **Label Ratio: 12.5%** | | | | |
> | TraPO (ID Avg.)  | 39.7 | 40.7 | 41.7 | 44.1 |
> | TraPO (OOD Avg.) | 47.6 | 52.1 | 54.2 | 56.1 |
> | **Label Ratio: 25%** | | | | |
> | TraPO (ID Avg.)  | 40.6 | 41.0 | 42.6 | 45.6 |
> | TraPO (OOD Avg.) | 48.5 | 55.7 | 56.1 | 59.7 |
> | **Label Ratio: 50%** | | | | |
> | TraPO (ID Avg.)  | 39.1 | 41.5 | 43.8 | 45.9 |
> | TraPO (OOD Avg.) | 51.2 | 54.1 | 55.7 | 59.2 |

---

> ### Author Response · Authors · 2025-11-21
>
> >**Q7:** Please include a sensitivity analysis for hyperparameters top-p and threshold  to better understand their influence on performance.
> >**A:**
> Thank you for your valuable suggestion. We have run sensitivity analyses on the key hyperparameters: top-p, $\Gamma$ , and warm-up length. We present the overall experimental results below. **For detailed results, please refer to Tables 9, 10 and 11 in the appendix of the updated paper.**
> For **top-p**, we find that larger values lead to noisier responses early in training, which degrade model performance due to unreliable pseudo-labels.
> For $\Gamma$, setting it too low (e.g., 0.1) admits too many low-quality unlabeled samples, while setting it too high (e.g., above 0.7) is overly conservative and underutilizes the unlabeled data. Both extremes reduce performance.
> As for **warm-up length**, a very short warm-up (e.g., 2 epochs) yields unreliable estimates when unlabeled samples are first introduced, hurting performance. Performance improves and stabilizes as warm-up length increases. We chose 5 epochs as a balance, long enough for stable pseudo-label estimation, yet early enough to start leveraging unlabeled data efficiently.
>
> | Top-p | 0.1 | 0.3 | 0.5 | 0.7 | 1.0 |
> |----------------|-----|-----|-----|-----|------|
> | TraPO (ID Avg.)  | 42.6 | 41.7 | 38.5 | 39.6 | 39.2 |
> | TraPO (OOD Avg.) | 56.1 | 53.6 | 50.5 | 52.0 | 50.2 |
>
> | $\Gamma$ | 0.1 | 0.3 | 0.5 | 0.7 | 1.0 |
> |----------------|-----|-----|-----|-----|------|
> | TraPO (ID Avg.)  | 39.4 | 41.9 | 42.6 | 39.6 | 39.8 |
> | TraPO (OOD Avg.) | 51.4 | 55.4 | 56.1 | 53.3 | 53.5 |
>
> | Warm-up Length | 2   | 3   | 5   | 8   | 12  |
> |-------------------------|-----|-----|-----|-----|-----|
> | TraPO (ID Avg.)  | 39.2 | 41.6 | 42.6 | 43.1 | 42.1 |
> | TraPO (OOD Avg.) | 51.1 | 54.3 | 56.1 | 56.6 | 55.6 |
>
> [1] Shao Z, Wang P, Zhu Q, et al. Deepseekmath: Pushing the limits of mathematical reasoning in open language models[J]. arXiv preprint arXiv:2402.03300, 2024.

---

> ### Author Response · Authors · 2025-11-25
>
> Dear Reviewer aiJH,
>
> We hope this message finds you well.
>
> Once again, we sincerely appreciate your valuable time and thoughtful feedback. We are reaching out to kindly remind you that, based on your suggestions, we have included additional experiments and discussions in our response.
> If you could take valuable time out of your schedule to review our reply, we would be greatly appreciative. Your timely feedback will help us confirm whether we have appropriately addressed the issues and suggestions you raised.
>
> Furthermore, your insights and professional expertise are crucial to us. If there are any further areas where we can enhance the quality of the paper, please let us know. We are more than willing to address any remaining concerns.
>
> Warmest regards,
> The Authors

---

### Official Review · Reviewer_UVjM · 2025-11-03

**Soundness:** 3
**Presentation:** 3
**Contribution:** 3
**Rating:** 4
**Confidence:** 3

**Summary:**

The paper proposes TRAPO, a semi-supervised variant of reinforcement learning with verifiable rewards (RLVR) designed to improve reasoning in large language models (LLMs) with minimal labeled data. The key idea is to use a small labeled dataset to guide policy optimization on much larger unlabeled corpora. TRAPO aligns unlabeled samples with labeled ones by comparing their pass rate trajectories—the evolution of correct-response rates during training—thereby selecting only reliable unlabeled examples for reward-based updates. This stabilizes learning and prevents the collapse common in unsupervised RLVR. Empirically, TRAPO achieves notable data efficiency: with only 1K labeled and 3K unlabeled examples, it outperforms fully unsupervised methods trained on 45K samples; with 4K labeled and 12K unlabeled, it surpasses even fully supervised models trained on all 45K labeled samples. The framework generalizes across domains and model architectures, offering a principled, efficient path for semi-supervised reasoning enhancement in LLMs

**Strengths:**

(1) My biggest takeaway is that carefully curated mixture of labeled and unlabeled data could improve the performance and solve the unstability issue raised in the unsupervised training.

(2) The paper provides rigorous theoretical analysis, including generalization and convergence proofs linking trajectory consistency with domain adaptation theory.

(3) The paper presentation is good and easy to follow, also comes with comprehensive baselines and benchmarks.

**Weaknesses:**

(1) My main concern is the practical usage of the method, to my understanding, the method works as a preprocessing data selection step before launching the training on mixture data, what would be the cost and running time for this step? if it needs a long trajectory to determine good samples, is this process even more costly than the rl training itself? Correct me if I am wrong

(2) I am not sure if the roll out pass rate is a 'stable' indicator of this problem, as everytime the roll out could be different, while the model weights keep updating and the selection mask is applied to question.

(3) I guess one baseline, where selecting samples using your methods vs random selection with the same amount, is missing, because increasing the number of unlabeled data could natuarally worsen the training

**Questions:**

(1) Is there a threshold that can adjust the total number of selected questions, i.e, sum of M (SigmaM), it could be better if there are trend curve and ablation that when SigmaM =  0, it shows the full grpo result with increment 0%, when it is 1, it gives marginal improvement like 0.6%, and when using your method, the increment is the optimal like 4.3%, and the performance at other possible values in between.

---

> ### Author Response · Authors · 2025-11-21
>
> **Response to Reviewer UVjM:**
> Thank you for your thoughtful comments and helpful suggestions! They have been incredibly valuable to our work. We apologize for the delay in our response; limited computational resources for LLM training mean that each experiment takes quite a while to complete. We are pleased, however, that we have been able to address most of the reviewers' questions positively.
> Below are our detailed responses to your comments:
>
>
> > **W1:**
> My main concern is the practical usage of the method, to my understanding, the method works as a preprocessing data selection step before launching the training on mixture data, what would be the cost and running time for this step? if it needs a long trajectory to determine good samples, is this process even more costly than the rl training itself? Correct me if I am wrong.
> >**A:**
> Thank you for raising this concern. In fact, TraPO is an **iterative semi-supervised reinforcement learning framework that operates throughout training** (as shown in Algorithm 1 of our paper), not a one-time preprocessing step. For ease of understanding, we have included a schematic diagram of the TraPO framework in the updated paper; **please refer to Figure 3**.
> At each step, TraPO seamlessly integrates an efficient data selection step into the training loop: (1) ```At each minibatch step```, it generates G responses per prompt using the current policy, as standard in RLVR; (2) during the usual reward evaluation, which is required by any RLVR method [1], **it simultaneously computes the (pseudo) pass rate** for each sample and updates its (pseudo) pass-rate trajectory for reliability assessment. Since trajectory matching involves only **simple statistics**, this incurs **negligible overhead**. (3) The policy is then updated on both labeled data and the currently reliable subset of unlabeled data via GRPO.
> We will further clarify the practical cost of TRAPO from two perspectives:
> **(1) Time complexity:**
> **Each labeled or unlabeled sample is rolled out $G$ times per epoch, in line with standard RLVR practices.** Let $T$ represent the total number of training epochs, $N_L, N_U$ the number of labeled and unlabeled samples, $ C_{\text{sim}}$ **the computational cost of a cosine similarity computation over short vectors** and $C_{\text{gen}}$ the computational cost of a single rollout.
> **The only additional operation is a cosine similarity computation $C_{\text{sim}}$ over short vectors, which is negligible compared to the cost of rollout generation $C_{\text{gen}}$, i.e, $C_{\text{sim}} << C_{\text{gen}}$.**
> The time complexity of fully supervised training (using $N = N_L + N_U$ labeled samples) is:
> $ T_{\text{Sup}} = O(T \cdot N \cdot G \cdot C_{\text{gen}})$
> TraPO has the same complexity:
> $T_{\text{TraPO}} = O\big(T \cdot (N_L + N_U) \cdot G \cdot C_{\text{gen}}\big) + O\big( T\cdot (N_L + N_U)\cdot C_{\text{sim}} \big)  \approx  O(T \cdot N \cdot G \cdot C_{\text{gen}})
> $
> Therefore, TRAPO and fully supervised RLVR share identical time complexity, both dominated by forward sampling and GRPO updates.
> **(2) Empirical training cost:**
> In our experiments, TRAPO, supervised RLVR, and unsupervised RLVR are trained with the same epochs, hardware (8xH200), and batch sizes. The time costs of each paradigm are shown in the table below. This shows that TraPO incurs no substantial additional training cost compared to supervised RLVR.
> Additionally, to explain why the effective duration of unsupervised training is so short: due to the excessive noise during unsupervised training, **the model quickly experiences model collapse**. We have included the time consumption comparison results in the appendix. Please refer to Table 7 in the updated paper.
>
> **Table: Wall-clock training time (reported as "GPU-hours × GPUs") across data regimes**
> | Data Size | Unsupervised (Early Collapse) | Supervised | Semi-Supervised (TraPO) |
> |-----------|--------------|------------|---------------------------|
> | 4K        | ~7 × 8       | ~25 × 8    | ~26 × 8                  |
> | 8K        | ~13 × 8      | ~39 × 8    | ~38 × 8                  |
> | 45K       | ~11 × 8      | ~57 × 8    | ~55 × 8                  |

---

> ### Author Response · Authors · 2025-11-21
>
> > **W2:** I am not sure if the roll out pass rate is a 'stable' indicator of this problem, as everytime the roll out could be different, while the model weights keep updating and the selection mask is applied to question.
> >**A:** Thank you for this thoughtful question. First, rollout instability typically occurs at the level of individual samples. However, our TRAPO approach mitigates this by determining the pseudo pass rate through voting across G independent rollouts, thereby enhancing stability. Previous work [2] also demonstrates similar robustness when prompting the LLM to generate multiple answers to reach a consensus result.
> Second, even if the pass rate for a particular epoch is not perfectly measured, this does not undermine TRAPO's effectiveness, as it does not rely on single-epoch pass rates. Instead, it monitors the empirical pass rate trajectory over multiple epochs for each unlabeled question and aligns it with the learning dynamics observed in labeled data, thereby further enhancing stability.
> The consistent gains in Tables 1-2 confirm that this signal is stable and effective in practice.
>
>
> > **W3:**  I guess one baseline, where selecting samples using your methods vs random selection with the same amount, is missing, because increasing the number of unlabeled data could natuarally worsen the training.
> >**A:** Thank you for these valuable suggestions, which have helped to make our experimental analysis more thorough. Specifically, we use the Qwen-2.5-7B model with 1K labeled and 3K unlabeled samples, training with varying proportions of unlabeled data selected randomly or via TraPO. Results appear in the table below.
> Through experiments, we observe that **TraPO outperforms random selection under multiple selection ratios**. Random selection confers little benefit at any ratio and often lowers performance, **underscoring the need for reliable sample selection**.
> Additionally, we observe that TraPO's performance improves as more samples are included up to a point (e.g, 30%), but declines if more are added. Beyond this point, the remaining samples are excessively noisy, so their inclusion diminishes rather than enhances performance. These observations demonstrate that **the closer a sample matches our metric, the higher the probability that its pseudo-label is correct**. For comprehensive results, please refer to Table 14 in the appendix. Figure 4 in the paper also demonstrates a positive correlation between the trajectory-matching pass rate and pseudo-label accuracy.
>
>
> | Selection Ratio | 0%   | 10%  | 30%  | 50%  | 70%  | 100% |
> |-----------------|------|------|------|------|------|-------|
> | Random (ID Avg.)     | 39.4 | 39.8 | 39.7 | 37.9 | 38.4 | 39.2  |
> | TraPO (ID Avg.)      | 39.4 | 40.7 | 41.8 | 40.1 | 39.8 | 39.2  |
> | Random (OOD Avg.)    | 52.1 | 52.5 | 53.3 | 51.5 | 52.3 | 50.2  |
> | TraPO (OOD Avg.)     | 52.1 | 53.6 | 55.5 | 54.0 | 53.1 | 50.2  |
>
>
> > **Q1:**  Is there a threshold that can adjust the total number of selected questions, i.e, sum of M (SigmaM), it could be better if there are trend curve and ablation that when SigmaM = 0, it shows the full grpo result with increment 0%, when it is 1, it gives marginal improvement like 0.6%, and when using your method, the increment is the optimal like 4.3%, and the performance at other possible values in between.
> >**A:**
> This is very insightful! Following your guidance, similarly, we use the Qwen-2.5-7B model with 1K labeled and 3K unlabeled samples. We train with varying ratios ($\sigma_M$) of unlabeled data (from the 3K pool) using two approaches. The first is the best-performing baseline, which combines token-level entropy with naive semi-supervised learning (which learns from both labeled and unlabeled data directly without data selection). The second is our TraPO method. Results are shown in the table below.
> **We find that TraPO shows a consistent performance improvement as usable unlabeled data increases**. In contrast, token-level entropy does not exhibit this trend and consistently underperforms TraPO. This highlights the importance of effective denoising and selection of unlabeled samples during training. For comprehensive results, please see Table 15 in the appendix.
>
> | $\sigma_M$       | 0%   | 25%  | 50%  | 75%  | 100% |
> |------------------------|------|------|------|------|------|
> | Token Entropy (ID Avg.)     | 39.4 | 40.4 | 38.6 | 38.9 | 40.0 |
> | TraPO (ID Avg.)             | 39.4 | 40.2 | 41.6 | 41.7 | 42.6 |
> | Token Entropy (OOD Avg.)    | 52.1 | 53.4 | 51.7 | 49.6 | 49.7 |
> | TraPO (OOD Avg.)            | 52.1 | 53.8 | 54.8 | 55.2 | 56.1 |
>
> [1] Shao Z, Wang P, Zhu Q, et al. Deepseekmath: Pushing the limits of mathematical reasoning in open language models[J]. arXiv preprint arXiv:2402.03300, 2024.
> [2] Wang X, Wei J, Schuurmans D, et al. Self-Consistency Improves Chain of Thought Reasoning in Language Models[C]//The Eleventh International Conference on Learning Representations.

---

> ### Author Response · Authors · 2025-11-25
>
> Dear Reviewer UVjM,
>
> We hope this message finds you well.
>
> Once again, we sincerely appreciate your valuable time and thoughtful feedback. We are reaching out to kindly remind you that, based on your suggestions, we have included additional experiments and discussions in our response.
> If you could take valuable time out of your schedule to review our reply, we would be greatly appreciative. Your timely feedback will help us confirm whether we have appropriately addressed the issues and suggestions you raised.
>
> Furthermore, your insights and professional expertise are crucial to us. If there are any further areas where we can enhance the quality of the paper, please let us know. We are more than willing to address any remaining concerns.
>
> Warmest regards,
> The Authors

---

### Author Response · Authors · 2025-12-01
**A summary of the key pros and concerns for the convenience of (senior) area chairs and reviewers**

Dear (Senior) ACs and Reviewers,

We extend our sincere gratitude for the time and effort you have dedicated to reviewing our manuscript. We highly value the thoughtful and constructive feedback from all reviewers, which has greatly contributed to improving the clarity, rigor, and overall quality of our work.

TraPO is **the first Semi-Supervised RLVR framework** that effectively **links labeled and unlabeled samples**, dramatically improving stability and preventing model collapse in low-label settings. With **only 10%** (4K) labels, it **outperforms a fully supervised model** trained with **45K** labels. We hope our work will inspire further research in this important area.


We appreciate the reviewers’ recognition of the novelty, theoretical grounding, and empirical rigor of our work. For your convenience, we summarize the key pros and concerns raised by the reviewers, alongside our responses:

> **Key pros noted by the reviewers:**
> **P1:** The method is **clear, intuitive, and well-motivated**, effectively leveraging labeled–unlabeled data and a simple trajectory-based mechanism (All Reviewers **UVjM**, **aiJH**, **NfpB**).
> **P2:** The paper provides **solid theoretical analysis**, including generalization and convergence insights tied to trajectory similarity (Reviewers **UVjM**, **NfpB**).
> **P3:** The **experiments are comprehensive and strong**, showing consistent ID/OOD gain and high label efficiency (All Reviewers **UVjM**, **aiJH**, **NfpB**).
> **P4:** The paper is **clearly written and well-organized**, making it easy to follow (Reviewers **UVjM**, **aiJH**).
> **P5:** The work addresses a **timely and practically relevant problem**, making the direction promising for the community (Reviewer **aiJH**).

> **Key concerns noted by the reviewers and our responses:**
> **C1:** _Training cost, feasibility, and scalability  of TraPO._
TraPO is **not** a pre-processing step but a lightweight module integrated into standard RLVR training and **linearly scalable** with data size (Reviewers **UVjM**, **aiJH**).
> **C2:** _Stability and reliability of pass-rate measurements._
While single-sample pass rates fluctuate, TraPO uses **majority voting** and, more importantly, the **trajectory trend across epochs**, which is highly stable. Experiments confirm that trajectory-based signals reliably distinguish learnable samples from noisy or degenerate ones (Reviewer **UVjM**).
> **C3:** _Why average trajectory? Why not clustering?_
Clustering is impractical for RLVR due to **high model cost** and **lack of shared semantics**.
Average learning trajectory is both **computationally efficient** and **semantically meaningful**, capturing key patterns—hard samples (flat near 0), overly easy samples (early spike), and learnable samples (steady growth) (Reviewer **aiJH**).
> **C4:** _Sensitivity to hyperparameters (top-p, $\Gamma$, warm-up)._
Experiments show:  **top-p** too large → noisy generation; moderate values best.  $\Gamma$ too low → noisy samples leak; too high → overly conservative.  **warm-up** < 5 epochs → unstable pseudo-labels; 5 is a good trade-off.  TraPO remains stable under reasonable ranges (Reviewers **aiJH**, **NfpB**).
>
> **C5:** _Comparison with random sampling / entropy / self-certainty baselines._
TraPO consistently outperforms random selection and prior confidence-based heuristics. Performance declines only when inclusion ratio becomes excessively high (>30%), confirming TraPO’s **noise-filtering ability** (Reviewers **UVjM**, **NfpB**).
> **C6:** _New dataset sizes and model variants._
We provided more comprehensive Llama-3.1-8B experiments than the initial version of the paper during the rebuttal, and TraPO remains SOTA.  Across label ratios (**12.5%/25%/50%**) and data scales (**1K/2K/4K/16K**), performance improves smoothly (Reviewer **aiJH**).

Thank you very much for your time and consideration. We hope this will be helpful!

Best regards,
Authors

---

### Meta-Review · Area_Chair_i1SN · 2026-01-09

**Summary:**

This paper proposes a new semi-supervised RLVR algorithm in training large reasoning models, TraPO (Trajectory-based Policy Optimization), which identifies reliable unlabeled samples by comparing their training trajectory to the labeled samples’. The reviewers raised several concerns in their reviews, mostly regarding insufficient experiments (e.g., hyperparameter sensitivity, diverse LLM family, additional baselines). The rebuttal improves experimental clarity and provides additional analyses.

**Reviewer Concerns:**

The summary of original concerns and how the rebuttal address them is as follow:

**UVjM (4, no response)**

- Concern regarding practical usage of TraPO => mostly due to misunderstanding by the reviewer
- Validity of majority-voted prediction as pseudo-label => supported by empirical effectiveness of TraPO and prior work
- Absence of comparison with random selection => addressed by new experiments in rebuttal

**aiJH (4, no response)**

- Insufficient empirical justification regarding some design choices => addressed by new experiments in rebuttal

**NfpB (4, response without changing score, but slightly positive)**

- Room to improve novelty positioning => addressed by new experiments in rebuttal
- Theory with simplified assumptions => clarification in rebuttal
- Insufficient experiments (sensitivity & cost, more non-math/OOD benchmarks) => additional experiments and clarifications in rebuttal

**Reviewer Scores:**

Initially, the reviewers' scores were (4,4,4), leaning to the rejection. Despite the authors providing the extensive rebuttal, only one reviewer (NfpB) responded to this with partial satisfaction yet no changed score (*“some of the concerns are addressed. I will wait until the end of the discussion period to make the final decision about my score.”*). After carefully reading all the reviews and rebuttals by the authors, AC believes that most of the concerns are now resolved and expect that all reviewers would have increased their scores to (6,6,6).

---

### Decision · Program_Chairs · 2026-01-26

Accept (Poster)